# FRET-based reporters for the direct visualization of abscisic acid concentration changes and distribution in Arabidopsis

Rainer Waadt[1,2], Kenichi Hitomi[3,4], Noriyuki Nishimura[1,2†], Chiharu Hitomi[3,4], Stephen R Adams[5], Elizabeth D Getzoff[3,4], Julian I Schroeder[1,2]*

[1]Division of Biological Sciences, Cell and Developmental Biology Section, University of California, San Diego, La Jolla, United States; [2]Food and Fuel for the 21st Century, University of California, San Diego, La Jolla, United States; [3]Department of Integrative Structural and Computational Biology, The Scripps Research Institute, La Jolla, United States; [4]The Skaggs Institute for Chemical Biology, The Scripps Research Institute, La Jolla, United States; [5]Department of Pharmacology, University of California, San Diego, La Jolla, United States

*For correspondence: jischroeder@ucsd.edu

Present address: †Institute of Radiation Breeding, National Institute of Agrobiological Sciences (NIAS), Kamimurata, Japan

Competing interests: The authors declare that no competing interests exist.

**Abstract** Abscisic acid (ABA) is a plant hormone that regulates plant growth and development and mediates abiotic stress responses. Direct cellular monitoring of dynamic ABA concentration changes in response to environmental cues is essential for understanding ABA action. We have developed ABAleons: ABA-specific optogenetic reporters that instantaneously convert the phytohormone-triggered interaction of ABA receptors with PP2C-type phosphatases to send a fluorescence resonance energy transfer (FRET) signal in response to ABA. We report the design, engineering and use of ABAleons with ABA affinities in the range of 100–600 nM to map ABA concentration changes in plant tissues with spatial and temporal resolution. High ABAleon expression can partially repress Arabidopsis ABA responses. ABAleons report ABA concentration differences in distinct cell types, ABA concentration increases in response to low humidity and NaCl in guard cells and to NaCl and osmotic stress in roots and ABA transport from the hypocotyl to the shoot and root.

## Introduction

Plant hormones control plant growth and development. Knowledge of the locations of hormone synthesis and transport, and the resulting hormone gradients and distributions in plants is important for understanding how plants respond to their environment via hormone signaling and cross talk of hormone and other small molecule signaling pathways (*Hetherington and Woodward, 2003*; *Israelsson et al., 2006*; *Nemhauser et al., 2006*; *Muday et al., 2012*). Among plant hormones, auxin is best characterized in terms of its distribution and transport (*Vanneste and Friml, 2009*), analyzed using reporter constructs for auxin-induced gene expression or protein degradation (*Ulmasov et al., 1997*; *Brunoud et al., 2012*; *Wend et al., 2013*). Reporter genes developed for ABA-induced gene expression (pRD29A/B, pRAB18 and pAtHB6) are also used (*Lång and Palva, 1992*; *Yamaguchi-Shinozaki and Shinozaki, 1993*; *Ishitani et al., 1997*; *Christmann et al., 2005*; *Kim et al., 2011*; *Duan et al., 2013*). Despite their potential, such promoter-linked reporters respond indirectly and slowly to their respective plant hormone. To unequivocally investigate dynamic models of hormone distribution and dissect the complex functions and interconnection of these signaling molecules, direct plant hormone reporters that act instantaneously and reversibly are essential.

Optogenetic reporters provide a potential solution. These genetically engineered chromogenic proteins (often fluorescent proteins) respond to a specific environmental change via conformationally

**eLife digest** Plants are able to respond to detrimental changes in their environment—when, for example, water becomes scarce or the soil becomes too salty—in ways that minimize stress and damage caused by these changes. Hormones are chemicals that trigger the plant's response under these circumstances.

Abscisic acid is the hormone that regulates how plants respond to drought and salt stress and that controls the plant growth in these conditions. In the past, it was possible to measure the average level of this hormone in a given tissue, but not the level in individual cells in a living plant. Moreover, it was difficult to follow directly how abscisic acid moved between the plant cells, tissues or organs.

Now, Waadt et al. (and independently Jones et al.) have developed tools that can measure the levels of abscisic acid within individual cells in living plants and in real time. The plants were genetically engineered to produce sensor proteins with two properties: they can bind to abscisic acid in a reversible manner, and they contain two 'tags' that fluoresce at different wavelengths. Shining light onto the plant at a specific wavelength that is only absorbed by one of the tags actually causes both of the tags on the sensor proteins to fluoresce. However, the sensors fluoresce more at one wavelength when they are bound to abscisic acid, and more at the other wavelength when they are not bound to abscisic acid. Hence, measuring the ratio of these two wavelengths in the light that is given off by the sensor proteins can be used as a measure of the concentration of abscisic acid in a plant cell.

Waadt et al. developed sensor proteins called 'ABAleons', and used one of these to analyze the uptake, distribution and movement of abscisic acid in different tissues in the model plant *Arabidopsis thaliana*. Changes in the level of abscisic acid could be detected at the level of an individual plant cell, and over time scales of fractions of seconds to hours. ABAleons also revealed that the concentration of abscisic acid in guard cells—specialized cells that help stop the loss of water vapor from a leaf—increases when humidity levels are low, or when salt levels are high. Low water levels, or high salt levels, also slowly increased the concentration of abscisic acid in the roots of the plant. Furthermore, Waadt et al. saw that abscisic acid moved long distances from the base of the stem up into the shoot, and down to the root.

Waadt et al. also report that the ABAleons made plants less responsive to abscisic acid, possibly because binding to the ABAleons reduced the amount of abscisic acid that was available to perform its role as a hormone. The next challenge is to engineer ABAleons that minimize this unwanted side effect, and then go on to use ABAleons to study environmental conditions and proteins involved in plant hormone responses.

linked changes in their spectral properties measurable with optical instruments (*Giepmans et al., 2006*; *Alford et al., 2013*). Such reporters have been developed for a whole palette of molecules and physiochemical processes (*Okumoto et al., 2012*). However, no reporters for direct visualization of any plant hormone have yet been developed.

During land colonization, plants adopted ABA as a hormone to signal stress due to limited water supply (*Cutler et al., 2010*; *Raghavendra et al., 2010*; *Hauser et al., 2011*). ABA is integrated into a complex signaling network that transcriptionally and post-translationally regulates seed germination, root development and stomatal aperture (*Hetherington and Woodward, 2003*; *Cutler et al., 2010*; *Kim et al., 2010*; *Raghavendra et al., 2010*; *Tanaka et al., 2013*). ABA biosynthesis is a multi-step reaction involving Zeaxanthin epoxidation, isomerization and cleavage to Xanthoxin in the plastid, followed by conversion to abscisic aldehyde and oxidization to ABA in the cytoplasm (*Nambara and Marion-Poll, 2005*). In Arabidopsis, ABA is synthesized primarily in vascular tissues of roots and leaves, in guard cells and in seeds (*Sauter et al., 2001*; *Endo et al., 2008*; *Seo and Koshiba, 2011*; *Bauer et al., 2013*; *Boursiac et al., 2013*). ABA catabolism includes hydroxylation pathways and glucose conjugation leading to less- or inactive compounds (*Nambara and Marion-Poll, 2005*; *Kepka et al., 2011*). ABA-glucose ester is stored in the vacuole and was reported to be rapidly hydrolyzed by β-glucosidases (*Lee et al., 2006*). However, direct measurements of rapid ABA release are missing.

ABA moves throughout the plant and crosses cell borders as a function of pH. This 'ionic trap model' explains the movement, but excludes cellular efflux of ABA due to low apoplastic pH (*Sauter et al., 2001*; *Seo and Koshiba, 2011*; *Boursiac et al., 2013*). Recently identified ABA transporters contribute to ABA export from the vasculature and import into guard cells (*Kang et al., 2010*; *Kuromori et al., 2010, 2011*; *Kanno et al., 2012*; *Boursiac et al., 2013*). Two non-mutually exclusive current models describe how water limitations in the root induce stomatal closure in the leaf: (1) ABA acts as a chemical signal synthesized in the root and transported to the shoot (*Sauter et al., 2001*; *Wilkinson and Davies, 2002*), and (2) a hydraulic signal from the root induces ABA synthesis in the shoot (*Christmann et al., 2005, 2007*; *Ikegami et al., 2009*).

In response to water limitations ABA concentrations increase (*Harris et al., 1988*; *Harris and Outlaw, 1991*; *Christmann et al., 2007*; *Forcat et al., 2008*; *Ikegami et al., 2009*; *Geng et al., 2013*) and decrease upon stress relief (*Harris and Outlaw, 1991*; *Endo et al., 2008*). Despite recent progress on ABA synthesis and transport, direct evidence for conditionally triggered changes in local ABA concentrations and time-resolved data for ABA re-distribution *in vivo* are lacking.

ABA is perceived by members of a protein family designated as PYRABACTIN RESISTANCE 1 (PYR1)/PYR1-LIKE (PYL)/REGULATORY COMPONENT OF ABA RECEPTOR (RCAR), which in the presence of ABA negatively regulate Clade A TYPE 2C PROTEIN PHOSPHATASES (PP2Cs) (*Ma et al., 2009*; *Park et al., 2009*). Inhibition of PP2Cs enables the activation of SNF1-RELATED KINASES 2 (SnRK2s) (*Fujii et al., 2009*; *Umezawa et al., 2009*; *Vlad et al., 2009*), that target transcription factors, ion channels and NADPH oxidases (*Kobayashi et al., 2005*; *Furihata et al., 2006*; *Geiger et al., 2009*; *Lee et al., 2009*; *Sato et al., 2009*; *Sirichandra et al., 2009, 2010*; *Brandt et al., 2012*).

PYR/PYL/RCARs contain an internal ligand-binding pocket flanked by two conserved loops, the 'Pro-cap/gate' and the 'Leu-lock/latch' (*Melcher et al., 2009*; *Nishimura et al., 2009*; *Santiago et al., 2009a*). ABA binding triggers these loops to rearrange, closing the lid over ABA. The conformational change and rearranged protein surface favors PP2C binding over receptor dimerization (*Melcher et al., 2009*; *Nishimura et al., 2009*; *Yin et al., 2009*). In the resultant ABA receptor–phosphatase complex, a conserved Trp residue from the PP2C inserts between the 'Pro-cap/gate' and the 'Leu-lock/latch' to further enclose the ABA (*Melcher et al., 2009*; *Miyazono et al., 2009*; *Yin et al., 2009*).

Here, we report the design, development and application of optogenetic FRET-based reporters for ABA (ABAleons), in which covalently attached PYR1 and the PP2C ABI1 are modulated upon ABA binding, triggering changes in fluorescence emission from attached fluorescent proteins. ABAleons can affect ABA responses at high concentrations and enable the analysis of time-dependent changes in ABA concentration, distribution and transport in live plants with appropriate resolution to monitor endogenous ABA concentration changes.

## Results

### ABAleon design and *in vitro* characterization

Based on structural analyses of PYR1 (*Nishimura et al., 2009*) and the PYL1-ABA-ABI1 complex (*Miyazono et al., 2009*) and a FRET cassette consisting of the fluorescent proteins mTurquoise (*Goedhart et al., 2010*) and Venus circularly permutated at amino acid 173 (cpVenus173; *Nagai et al., 2004*; *Piljić et al., 2011*), FRET-based reporters for ABA, named ABAleons, were designed (*Figure 1A*). Full length PYR1 and ABI1 truncated at amino acid S125 ($_{\Delta N}$ABI1) were fused via a flexible ASGGSGGTS(GGGGS)$_4$-linker (*Arai et al., 2004*; *Nagai et al., 2004*) and inserted into the mTurquoise-cpVenus173 FRET cassette using short two amino acid GP- and PG-linkers (*Piljić et al., 2011*) resulting in the FRET reporter ABAleon1.1 (*Figure 1A*). Due to the long flexible linker between PYR1 and $_{\Delta N}$ABI1 (> 120 Å; *Arai et al., 2004*; *Figure 1A*), $_{\Delta N}$ABI1 might be free for substrate access in the ABA unbound conformation. Therefore the wild type ABAleon1.1 was mutated to abolish phosphatase activity of ABI1 by introducing a D413L mutation in the catalytic metal-binding site (ABAleon2.1; http://www.uniprot.org/uniprot/P49597) (*Figure 1A*).

*In vitro* application of ABA had no impact on ABAleon2.1 absorbance (*Figure 1—figure supplement 1A*). However, analyses of fluorescence emission spectra after application of ABA revealed an increase in mTurquoise emission (peak at 476 nm) and a decrease of cpVenus173 emission (peak at 527 nm), indicating that the distance between both fluorescent proteins is increased, or their orientation to each other is changed by the ABA-dependent interaction of PYR1 and $_{\Delta N}$ABI1 (*Figure 1B–D*). Apparent ABA affinities were calculated by curve fitting of ABA-dependent emission ratio plots

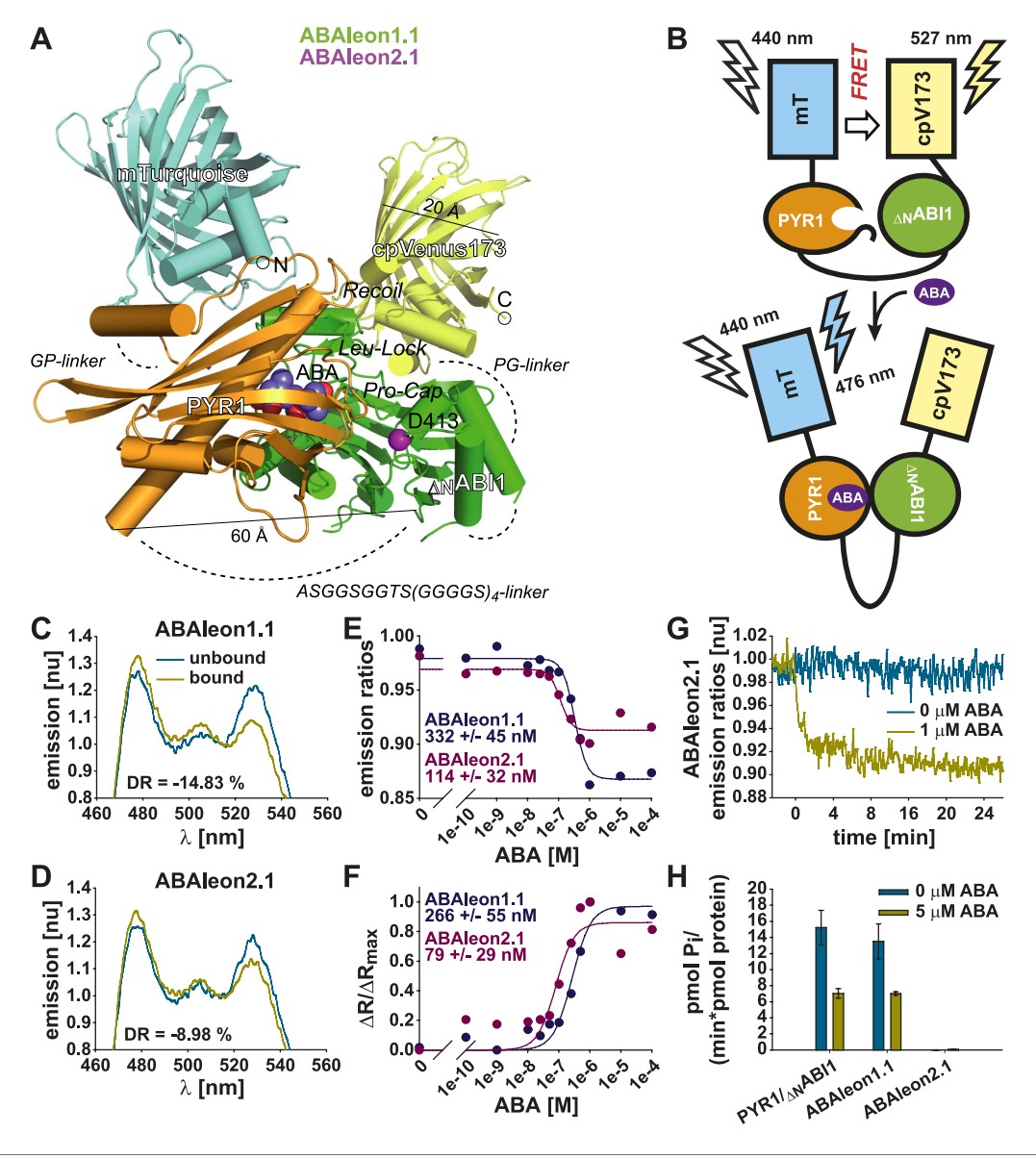

**Figure 1**. *In vitro* characterization of ABAleons. (**A**) mTurquoise (cyan) is fused through a GP-linker to PYR1 (gold), which is separated by a flexible ASGGSGGTS(GGGGS)₄ linker from ₐₙABI1 (green) fused to cpVenus173 (yellow) through a PG linker. Structural features of the PYR1-ₐₙABI1 complex including ABA (blue and red balls), ABI1 D413 (purple ball) and loops controlling access to the ABA binding site are highlighted. Dashed lines indicate linkers and unresolved structures. (**B**) Without ABA, ABAleon flexibility enables FRET from mTurquoise (mT) to cpVenus173 (cpV173). ABA triggered PYR1-ₐₙABI1 binding increases the distance or orientation between the fluorescent probes, thereby reducing FRET efficiency. (**C** and **D**) Normalized (nu) emission spectra of (**C**) ABAleon1.1 and (**D**) ABAleon2.1 in absence (unbound) and in presence (bound) of ABA with indicated dynamic range (DR). (**E**) Emission ratios and (**F**) ΔR/ΔRmax plotted against increasing [ABA], with indicated ABA affinity K′d of each ABAleon calculated from the respective plot. (**G**) Time-dependent normalized emission ratios of ABAleon2.1 in response to 0 and 1 μM ABA. (**H**) Phosphatase activity assays of equimolar PYR1 and ₐₙABI1 combinations and indicated ABAleons in presence of 0 and 5 μM ABA (mean ± SD, n = 4).

The following figure supplements are available for figure 1:

**Figure supplement 1**. ABA does not affect ABAleon absorbance and ABAleon emission is stable at physiological pH conditions.

(*Figure 1E*) or from fits of the ABA-dependent ratio change (ΔR) relative to the maximum ratio change (ΔR$_{max}$) (*Figure 1F*). For ABAleon1.1 a dynamic range of ~ 15 % was recorded with an apparent ABA affinity K'$_d$ of ~ 300 nM (*Figure 1C,E,F*). Moreover, ABA-induced emission ratio changes of ABAleon1.1 were stable in the range of physiological pH conditions (e.g., pH 6.6–8.2; *Figure 1—figure supplement 1B*). ABAleon2.1 exhibited a dynamic range of ~ 9 % and a K'$_d$ of ~ 100 nM (*Figure 1D–F*). In plate reader-based analyses, application of ABA rapidly and clearly induced emission ratio changes of ~ 8 % when analyzing ABAleon2.1 at 1 µM ABA (*Figure 1G*). ABAleon1.1 exhibited phosphatase activity comparable to PYR1 and $_{ΔN}$ABI1 when combined in a 1:1 molar ratio (*Figure 1H*). In the presence of 5 µM ABA phosphatase activity was inhibited to 50 % of initial activity (*Figure 1H*; *Ma et al., 2009*; *Park et al., 2009*; *Santiago et al., 2009b*). However, the predicted phosphatase inactive ABI1$_{D413L}$ mutation in ABAleon2.1 enabled the design of an ABA-reporter without detectable phosphatase activity (*Figure 1H*), which was considered to be preferable for use in plants.

## Application of ABA induces ABAleon2.1 emission ratio changes in Arabidopsis

ABAleon2.1 was transformed into the Arabidopsis Columbia 0 accession (Col-0) to determine whether it can detect ABA level changes *in planta*. In a macroscopic view ABAleon2.1 emission ratio maps were recorded from 5 day-old seedlings before (*Figure 2A*) or 2 h after ABA application (*Figure 2B*). Emission ratios were also recorded from guard cells of 33-day-old soil-grown plants (*Figure 2C*). As indicated in the color code of the calibration bar (*Figure 2*), low ABAleon2.1 ratios (blue) indicate high ABA concentrations and high ratios (red) indicate low ABA concentrations. Comparison of the emission ratio maps before and after ABA application revealed visible ABA uptake into the entire seedling (*Figure 2A,B*). However the most prominent ratio changes were observed in the root elongation- and early maturation zone, where the yellow-coded regions showed an ABA concentration increase, which is indicated by the yellow-to-blue color shift (*Figure 2A,B*). Visible ratio changes were also observed in the lower hypocotyl. Here an upward-directed ABA accumulation was detected (*Figure 2A,B*).

ABAleon2.1 emission ratios in guard cells were low (*Figure 2C*, blue), indicating elevated ABA concentrations under the imposed conditions. High ABA concentrations in guard cells were consistent with the constitutive guard cell expression of the ABA-induced reporter pRAB18-GFP when plants were grown at 70 % relative humidity (*Figure 2—figure supplement 1A*). Expression of pRAB18-GFP was further induced by ABA (*Figure 2—figure supplement 1B*), however with stronger induction 2 days after transfer to 25 % relative humidity (*Figure 2—figure supplement 1C,E*).

ABA-induced ABAleon2.1 responses were then analyzed with higher time- and spatial resolution in guard cells of 45-day-old plants (*Figure 3A–C*) and in three differentially color-coded regions of the hypocotyl (*Figure 3D–F*), root differentiation–(*Figure 3G–I*), root maturation–(*Figure 3J–L*) and root elongation-zone (*Figure 3M–O*) of 5-day-old seedlings. Application of ABA induced increases in mTurquoise (*Figure 3A,D,G,J,M*, mT, solid lines) and decreases in cpVenus173 emission (*Figure 3D,G,J,M*, cpV, dashed lines), resulting in an up to 12 % decrease in the emission ratios (*Figure 3B,E,H,K,N*). The ABA-induced ABAleon2.1 emission ratio changes indicate increases in the ABA concentration in all investigated Arabidopsis tissues. These *in planta* analyses were consistent with *in vitro* analyses (*Figure 1E,G*). The ABA-induced ABAleon2.1 ratio changes in guard cells (3–6 %; *Figure 3B*) and in the root differentiation zone (6 %; *Figure 3H*) were low compared to changes in the hypocotyl and lower root tissues (9–12 %; *Figure 3E,K,N*), consistent with data indicating elevated ABA concentrations prior to ABA application (*Figure 2A,C*). While ABA uptake into guard cells (*Figure 3B*) and into the root occurred simultaneously (*Video 1*) in all three analyzed regions (*Figure 3H,K,N*), a delay in the ABAleon2.1 response was observed in the hypocotyl (*Figure 3E*; *Video 1*). These data suggest a directional 'wave-like' ABA transport in the hypocotyl, which was also indicated in the ratio images (*Figure 3F*, *Video 1*). To describe the ABA transport in the hypocotyl more quantitatively, ABAleon2.1 response curves of all three analyzed regions were fitted by a four parameter logistic curve $R = R_{min} + \frac{R_{max} - R_{min}}{1 + (\frac{t}{t_{1/2}})^n}$. From these fits the t$_{1/2}$ values measure the time point when ABAleon2.1 is half-saturated. These values were used to calculate the delay in ABAleon2.1 responses between the analyzed regions which is a measure for the speed of ABA transport. From three independent experiments the rate of ABA transport in the hypocotyl was calculated as 16.4 ± 0.8 µm/min.

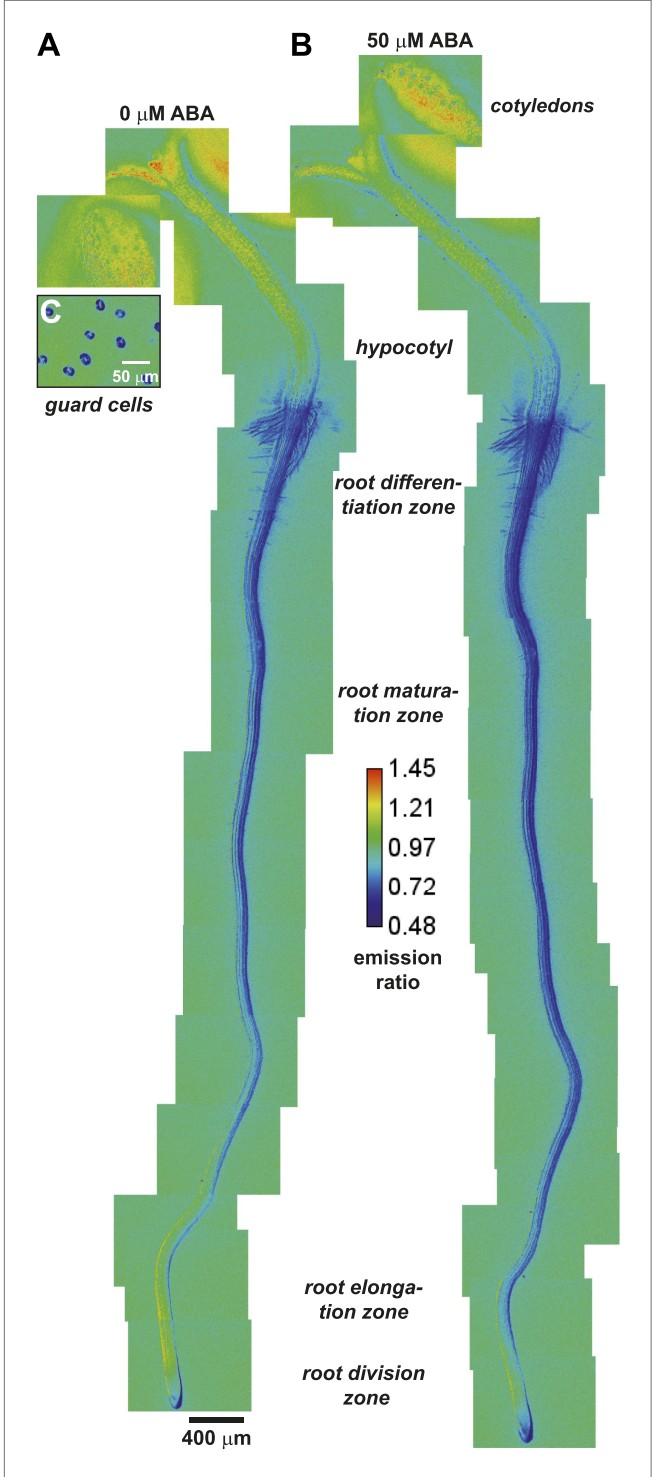

**Figure 2**. ABA-induced ABAleon2.1 responses in whole seedlings. (**A** and **B**) Manually assembled ABAleon2.1 ratio images (**A**) before and (**B**) 2 h after application of 50 µM ABA. (**C**) Ratio image of untreated guard cells from lower epidermis of 33-day-old soil grown plants. Images were calibrated to the indicated calibration bar. Blue colors indicate low ABAleon2.1 emission ratios, corresponding to high ABA concentrations, and red colors indicate high ABAleon2.1 emission ratios corresponding to low ABA concentrations. Shown is a representative of four experiments.

The following figure supplements are available for figure 2:

**Figure supplement 1**. pRAB18-GFP expression in guard cells.

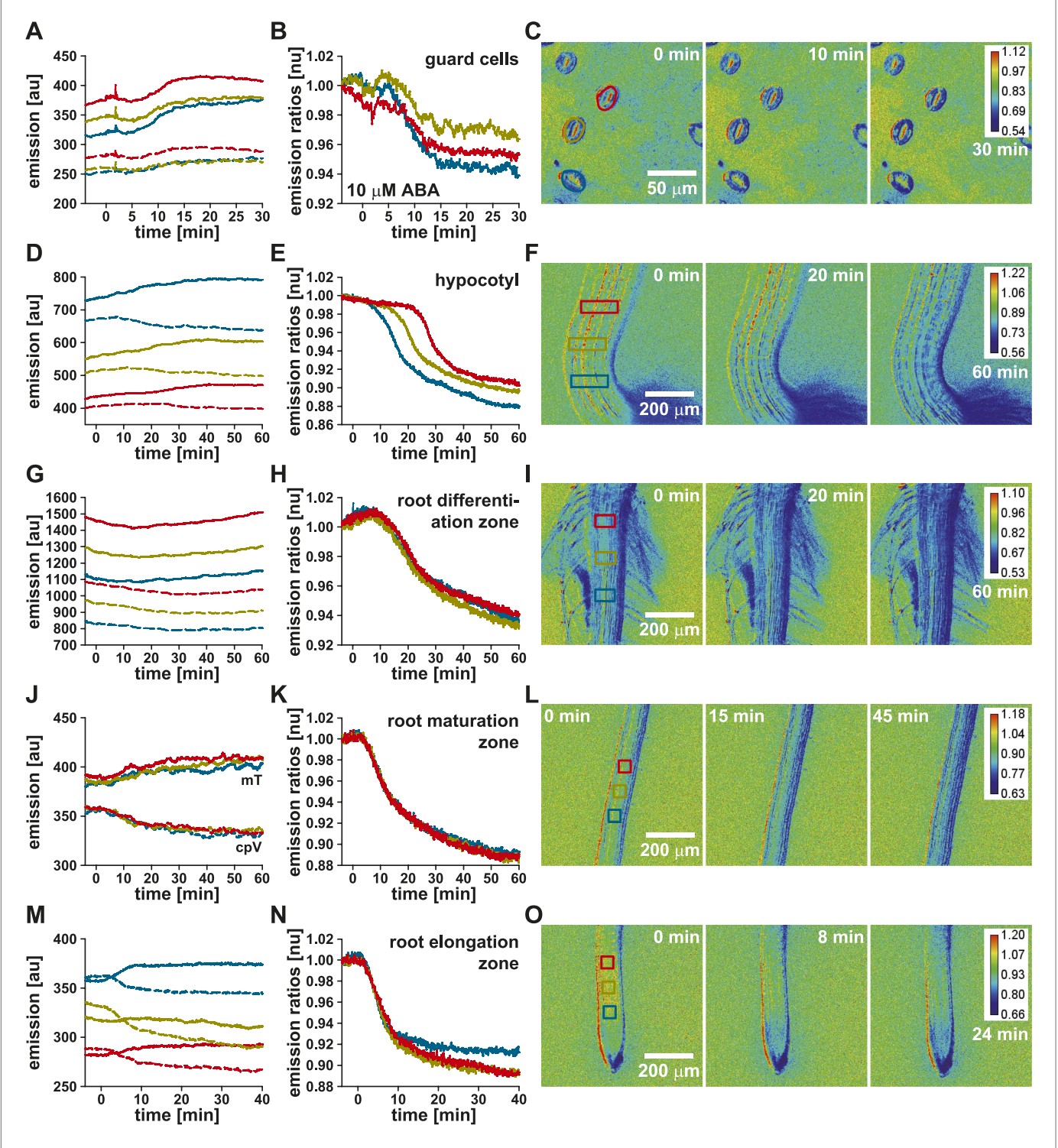

**Figure 3**. ABA-induced ABAleon2.1 responses in Arabidopsis tissues. Time-resolved ABAleon2.1 responses to 10 µM ABA in (**A–C**) guard cells of 45-day-old plants and (**D–F**) the hypocotyl, (**G–I**) the root differentiation-, (**J–L**) maturation- and (**M–O**) elongation-zone of 5-day-old seedlings. (**A, D, G, J, M**) Time course of mTurquoise (mT, solid lines) and cpVenus173 emission (cpV, dashed lines) and (**B, E, H, K, N**) the corresponding normalized emission ratios colored according to the analyzed regions boxed in the in initial t = 0 min images (**C, F, I, L, O**). Each analysis is a representative of 3–4 experiments. Note, that there is a slight sample drift, which causes cpVenus173 emission increases in (**A**).

*Figure 3. Continued on next page*

*Figure 3. Continued*

The following figure supplements are available for figure 3:

**Figure supplement 1**. ABAleon2.1 but not the empty FRET cassette responds specifically to ABA.

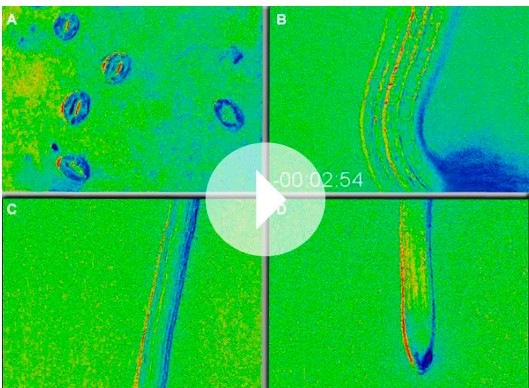

**Video 1**. 10 µM ABA-induced ABAleon2.1 responses in Arabidopsis. Video of 10 µM ABA-induced ABAleon2.1 responses in (**A**) guard cells, (**B**) the hypocotyl, (**C**) the root maturation- and (**D**) elongation-zone. ABA was applied at timepoint 00:00:00 of the indicated timescale. Single videos represent data of analyses in *Figure 3*. Emission ratio changes to blue color indicate ABA concentration increase. (**B**) Emission ratio changes in the hypocotyl propagate gradually from the hypocotyl base towards the shoot.

As ABA was dissolved in EtOH, responses to EtOH as solvent control were analyzed in the hypocotyl (*Figure 3—figure supplement 1A–C*) and the root maturation zone (*Figure 3—figure supplement 1D–F*). These treatments did not induce measurable emission ratio changes of ABAleon2.1 (*Figure 3—figure supplement 1B,E*). In contrast, subsequent application of ABA induced ABAleon2.1 emission ratio changes similar to previous data (*Figure 3E,K*, *Figure 3—figure supplement 1B,E*). In control seedlings expressing only the empty FRET-cassette no response to ABA was detected (*Figure 3—figure supplement 1G–I*). These data support, that PYR1-$_{ΔN}$ABI1$_{D413L}$ incorporated in ABAleon2.1 (*Figure 1A*) are responsible for the ABA-induced emission ratio changes in Arabidopsis. Taken together, these data clearly demonstrate the direct and instantaneous ABA detection by ABAleon2.1 in various tissues and the visualization of ABA transport *in planta*.

## Accelerated ABA-induced ABAleon2.1 responses in roots of an ABA receptor quadruple mutant

ABAleon2.1 was transformed into a PYR/PYL/RCAR ABA receptor quadruple mutant *pyr1-1/pyl1-1/pyl2-1/pyl4-1* (*pyl4ple*; *Park et al., 2009*; *Nishimura et al., 2010*). ABA response curves of Col-0 wild type (*Figure 4A*) and *pyl4ple* (*Figure 4B*) were fitted by a four parameter logistic curve using data of four single measurements (*Figure 4A,B*) or the combined datasets (*Figure 4C*). Data show that in the *pyl4ple* mutant ABAleon2.1 exhibited a faster response to 10 µM ABA in the root maturation zone when compared to Col-0 wild type (*Figure 4A–C*). This finding is also reflected in the t$_{1/2}$ values (*Figure 4D*). Half saturation of ABAleon2.1 was reached 16 min after ABA application in Col-0, while this appeared within 7 min in the *pyl4ple* mutant (*Figure 4C,D*).

## Arabidopsis plants expressing ABAleon2.1 are ABA hyposensitive

To investigate whether ABAleon2.1 might affect ABA responses in general, two ABAleon2.1 lines (line 3 and line 10) were compared to Col-0 wild type, YFP-PYR1 and *abi1-3*/YFP-ABI1 (*Nishimura et al., 2010*) over-expression lines. Analyses of the cpVenus173/YFP fluorescence emissions of the investigated lines indicated, that ABAleon2.1 (line 3) exhibited a ~ fivefold higher fluorescence emission (expression) when compared to ABAleon2.1 (line 10) (*Figure 5A*) while emission of YFP-ABI1 was ~ 7 % compared to YFP-PYR1 (*Figure 5A*).

In seed germination (*Figure 5F,G*) and cotyledon expansion assays (*Figure 5B,C,H*) ABAleon2.1 lines were hyposensitive to 0.8 µM ABA when compared to Col-0 wild type and YFP-PYR1. However, the ABAleon2.1 lines exhibited a less pronounced ABA hypersensitivity when compared to *abi1-3*/YFP-ABI1 (*Figure 5G,H*). Interestingly, the degree of ABA hyposensitivity of both ABAleon2.1 lines correlated with ABAleon2.1 expression levels (fluorescence emission), as the stronger expressing line 3 exhibited a reduced ABA sensitivity when compared to the lower expressing line 10 (*Figure 5*).

In seedling assays 10 µM ABA inhibited growth of all investigated lines (*Figure 5D,E,I*). Fresh weight of ABAleon2.1 and *abi1-3*/YFP-ABI1 plants, when grown on media supplemented with 10 µM

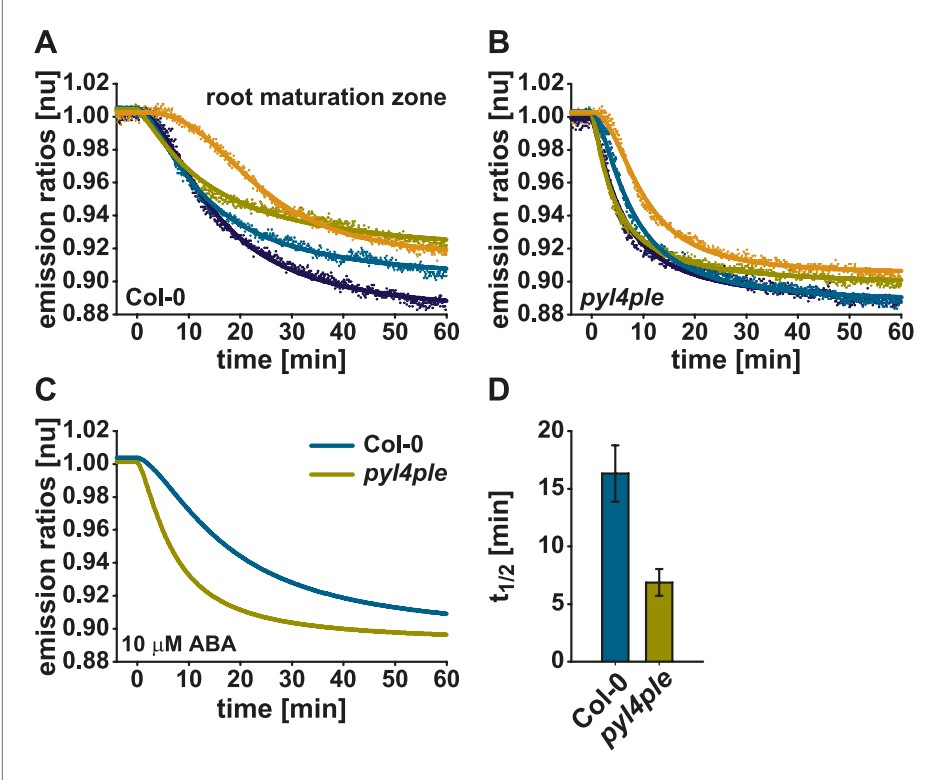

**Figure 4**. Accelerated ABAleon2.1 responses in roots of the *pyl4ple* mutant. Normalized 10 µM ABA-induced ABAleon2.1 emission ratio changes in the root maturation zone of Col-0 (**A**, **C** cyan line) and *pyr1-1/pyl1-1/pyl2-1/ pyl4-1* (*pyl4ple*) (**B**, **C** yellow line). (**A** and **B**) Data points from single measurements fitted by the respective four parameter logistic curve. (**C**) Combined data from four experiments in (**A** and **B**) fitted by the respective four parameter logistic curve. (**D**) $t_{1/2}$ values (means ± SEM, n = 4) calculated from the fitted curves in (**A** and **B**).

ABA, was less reduced compared to Col-0 wild type (*Figure 5I*). Again, the degree of ABA sensitivity of the ABAleon2.1 lines correlated with ABAleon2.1 expression levels (*Figure 5A,I*). Interestingly, growth and root length of YFP-PYR1 plants was drastically reduced, when grown on 10 µM ABA media (*Figure 5D,E,I*), suggesting a strong ABA hypersensitivity when over-expressing this receptor.

In ABA-induced stomatal closure assays ABA sensitivity of ABAleon2.1 lines was comparable to Col-0 wild type responses (*Figure 5J*), suggesting potential tissue or cell specific effects of ABAleon2.1 on ABA responses, which may be linked to a higher basal ABA concentration in these cells (*Figure 2C*, *Figure 2—figure supplement 1*). Taken together, ABAleon2.1 plants exhibit a reduced ABA sensitivity in seed germination and seedling growth which correlated with ABAleon2.1 expression levels.

## Visualization of long-distance ABA transport

To study long-distance ABA transport, modeling clay was placed into the middle of each imaging chamber to generate two isolated chambers (*Figure 6A*). Seedlings were placed such that the hypocotyl base and root differentiation zone laid over the isolating modeling clay, which isolated the shoot (top chamber) from the root (bottom chamber). Ratio images of the shoot/upper hypocotyl (*Figure 6E*) and the root maturation zone (*Figure 6F*) were recorded before and after application of 50 µM ABA. Three regions in the upper hypocotyl and the root maturation zone, indicated by boxes in the t = 0 min ratio images (*Figure 6E,F*), were used to measure time dependent ratio changes (*Figure 6B*). Upon ABA application to the upper chamber, ABA concentrations increased in the shoot/upper hypocotyl, as seen by an immediate decrease of the ABAleon2.1 emission ratios (*Figure 6B,E*). In the root maturation zone a decrease in the emission ratio was observed starting ~ 90 min after the treatment (*Figure 6B,F*).

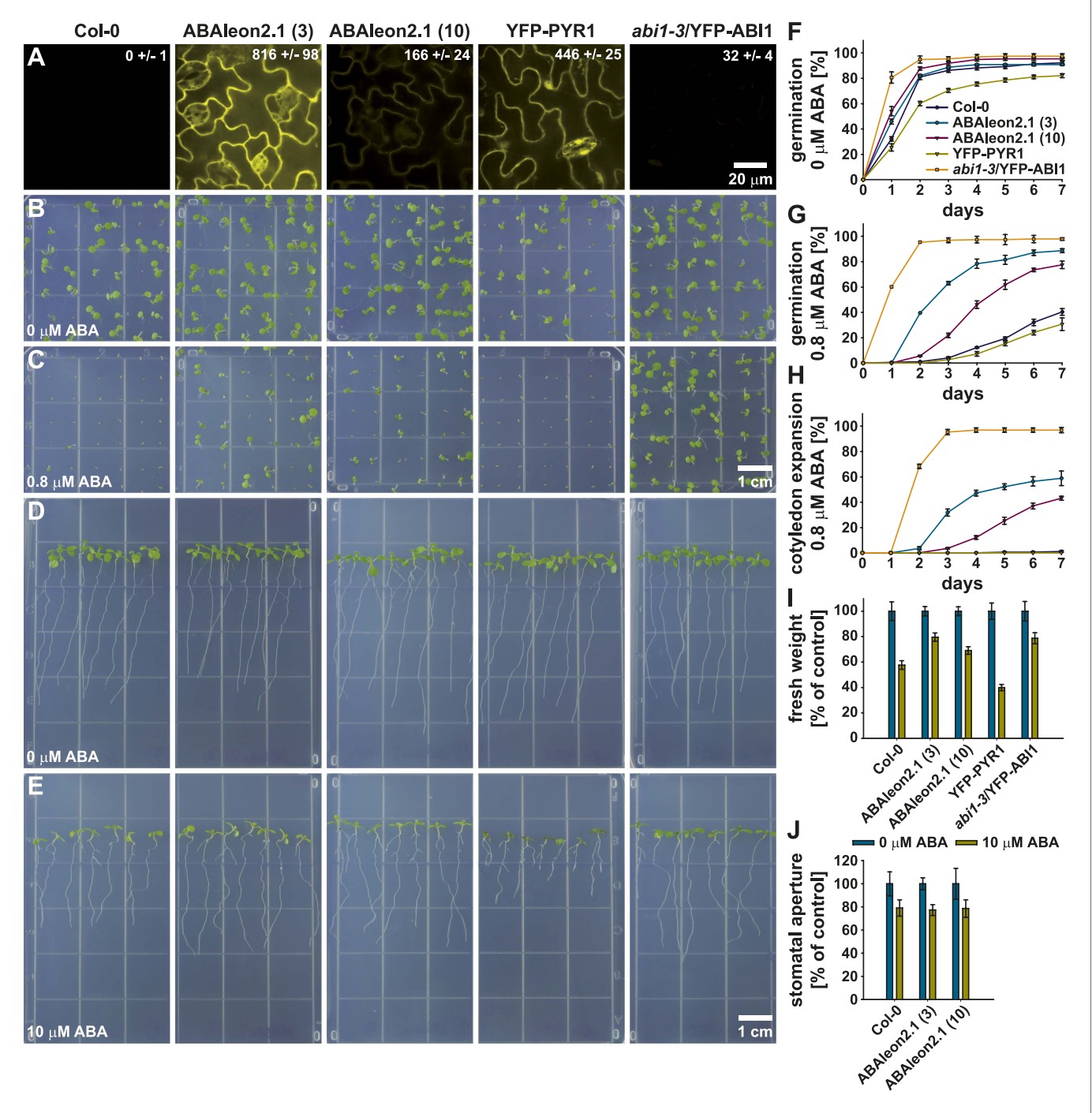

**Figure 5**. ABAleon2.1-expressing plants show an ABA hyposensitivity. From left to right, Col-0 wild type, ABAleon2.1 (line 3), ABAleon2.1 (line 10), YFP-PYR1 and *abi1-3*/YFP-ABI1. (**A**) Analyses of cpVenus173/YFP fluorescence emission in the leaf epidermis. Numerical fluorescence intensity values in the images represent means ± SEM of n = 4 images. (**B** and **C**) 7-day-old seedlings germinated and grown on 0.5 MS media supplemented with (**B**) 0 and (**C**) 0.8 μM ABA. (**D** and **E**) 9-day-old seedlings 5 days after transfer to 0.5 MS media supplemented with (**D**) 0 and (**E**) 10 μM ABA. (**F**–**H**) 7 day time course of (**F** and **G**) seed germination and (**H**) cotyledon expansion in presence of (**F**) 0 and (**G** and **H**) 0.8 μM ABA normalized to the seed count of each replicate (means ± SEM, n = 4 technical replicates with 49 seeds/n). (**I**) Fresh weight of seedlings from (**D** and **E**) normalized to the 0 μM ABA control conditions (means ± SEM, n = 5 technical replicates with seven seedlings/n). (**J**) Stomatal aperture of 20-23-day old seedlings exposed to 10 μM ABA normalized to the 0 μM ABA control conditions (means ± SEM, n = 3 with ≥ 24 stomata/n).

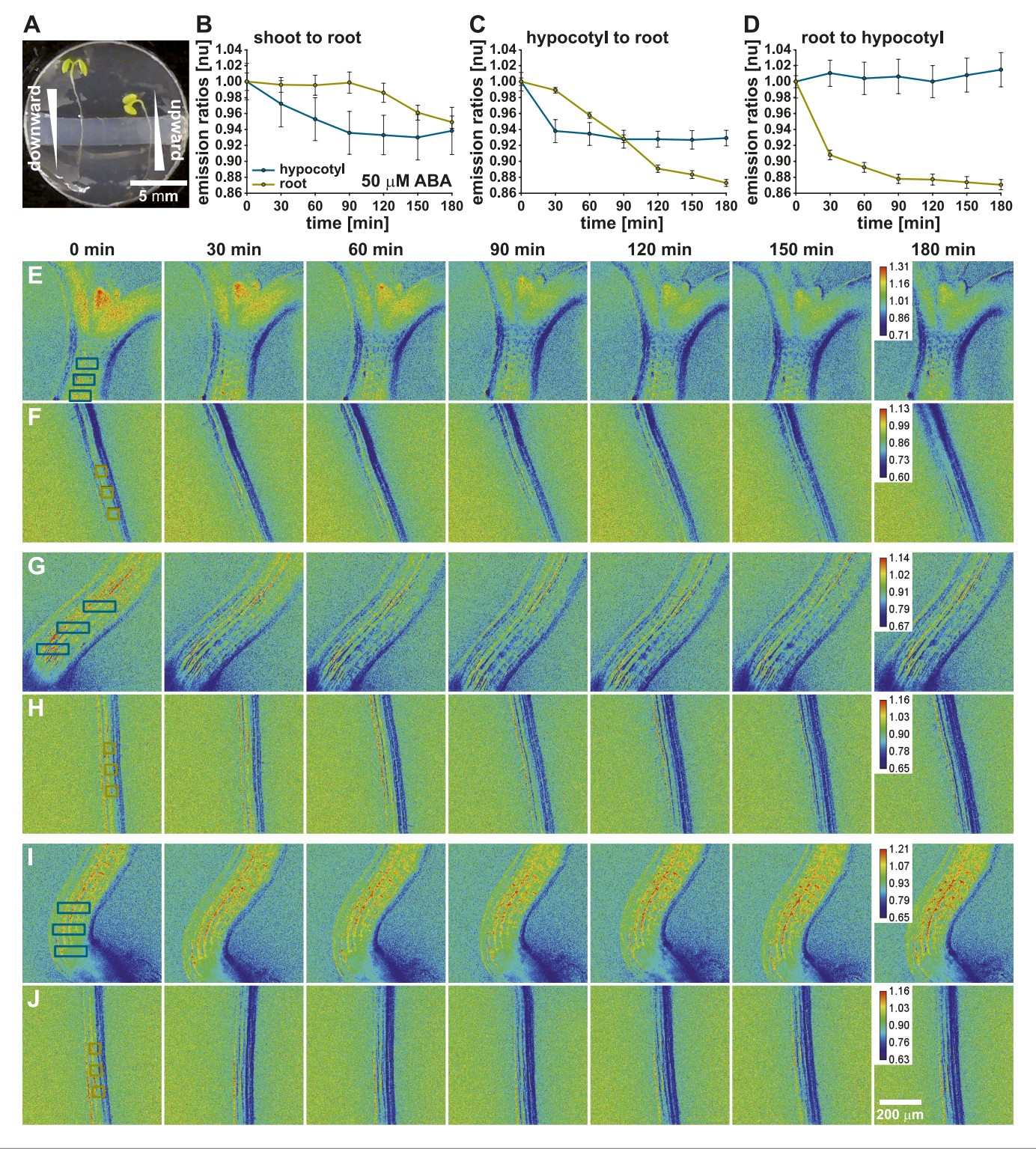

**Figure 6**. Visualization of long-distance ABA transport. (**A**) ABAleon2.1 seedlings were transferred to microscope dishes, which were divided into two isolated experimental chambers by a horizontal block of modeling clay. (**B**, **E**, **F**) Shoot-to-root, (**C**, **G**, **H**) hypocotyl-to-root and (**D**, **I**, **J**) root-to-hypocotyl ABA transport after application of 50 μM ABA. (**B–D**) Time-dependent normalized emission ratios (means ± SEM, n = 3) in the hypocotyl (cyan) and root (yellow) were quantified in three regions indicated by boxes in the initial images (**E–J**). The calibration bar in the final t = 180 min image indicates the scale of the emission ratios. Decreasing ratios indicate ABA accumulation. Shown are representative analyses of 3–4 experiments.
*Figure 6. Continued on next page*

eLIFE Research article

*Figure 6. Continued*

The following figure supplements are available for figure 6:

**Figure supplement 1**. Solvent control for long-distance ABA transport.

In additional experiments seedlings were placed such that the hypocotyl and root differentiation zone were located in the top chamber and the root maturation zone in the bottom chamber (*Figure 6A*, left seedling). In control experiments where 0.05 % EtOH as solvent control was added to the top chamber, no emission ratio changes were detected in the hypocotyl and the root maturation zone, indicating that in these experimental conditions endogenous ABA concentrations were stable (*Figure 6—figure supplement 1*). However, ABA application to the top chamber induced an ABAleon2.1 emission ratio change in the hypocotyl after 30 min which did not drastically change for the remaining time period (*Figure 6C,G*). ABA application to the hypocotyl (top chamber) led to a gradual decrease of the ABAleon2.1 emission ratios in the root maturation zone (bottom chamber; *Figure 6C,H*), providing evidence that ABA is actively transported to the root maturation zone. When ABA was applied to the root maturation zone (bottom chamber), ABAleon2.1 emission ratios rapidly dropped, indicating ABA uptake into the root (*Figure 6D,J*). However, ABA was not transported upwards to the hypocotyl (top chamber) within 180 min under the imposed conditions (*Figure 6D,I*). The above data indicate a shoot to root ABA transport (*Figure 6B,C*). A root to shoot ABA transport could not be detected within 180 min after ABA application (*Figure 6D*) possibly due to low transpiration when whole seedlings were perfused with buffer solution ('Materials and methods and Discussion').

## Mutations in PYR1 modulate ABA affinity and stereospecificity of ABAleons

Based on structural models (*Figure 7A*, *Figure 7—figure supplement 1A*), mutations in PYR1 and $_{\Delta N}$ABI1$_{D413L}$ of ABAleon2.1 were selected that potentially reduce but not abolish PYR1-$_{\Delta N}$ABI1$_{D413L}$ interaction (*Melcher et al., 2009*; *Miyazono et al., 2009*; *Mosquna et al., 2011*; *Zhang et al., 2012*) resulting in the new constructs ABAleon2.11–ABAleon2.17 (*Figure 7—figure supplement 1A*; *Table 1*). In addition shorter linker versions between PYR1 and $_{\Delta N}$ABI1$_{D413L}$ were generated (ABAleon2.2 and ABAleon2.3; *Table 1*). Recombinant empty FRET control (F3) and all ABAleon versions were purified (*Figure 7—figure supplement 2*) and biochemical characteristics of these proteins are provided in *Table 1*.

Analyses showed that linker deletions slightly changed ABA affinity but also reduced the dynamic range (*Table 1*). Compared to other investigated ABAleon2.1 mutants, PYR1$_{H60P}$ in ABAleon2.11 strongly impaired ABA-induced emission changes (*Figure 7B*), which were comparable to ABA-bound ABAleon2.1 (*Figure 7C*; *Table 1*). Of the seven analyzed ABAleon2.1 mutants, PYR1$_{V83H}$ in ABAleon2.13 and PYR1$_{H115A}$ in ABAleon2.15 were of particular interest, as these mutants exhibited a reduced ABA affinity (K'$_d$ ~ 3–4 µM of ABAleon2.13 and ~ 500–600 nM of ABAleon2.15; *Figure 7E*; *Table 1*). Also the dynamic range of these ABAleons was not drastically affected (*Figure 7D,F*; *Table 1*).

Unless otherwise stated, all of the above analyses have been conducted using the natural (+)-enantiomer of ABA. In additional experiments, apparent affinities for (+)-ABA (*Figure 7E*, *Figure 7—figure supplement 1B–D*) were compared to binding of its unnatural enantiomer (−)–ABA (*Figure 7G*, *Figure 7—figure supplement 1E–G*). In these analyses the dynamic ranges and (+)–ABA affinities of ABAleon1.1 (K'$_d$ ~ 300 nM) and ABAleon2.1 (K'$_d$ ~ 100 nM) (*Figure 7—figure supplement 1B–D*) were comparable to previous analyses (*Figure 1E,F*; *Table 1*). However, the binding affinity for (−)–ABA was reduced by about twofold in ABAleon1.1 (K'$_d$ ~ 600 nM) and ABAleon2.1 (K'$_d$ ~ 180 nM) (*Figure 7—figure supplement 1E–G*). In case of ABAleon2.13, affinities for (+)- and (−)-ABA were comparably low (K'$_d$ ~ 3–4 µM) and ABAleon2.13 did not reach saturating conditions at 200 µM (+)- or (−)-ABA (*Figure 7E,G*, *Figure 7—figure supplement 1B–G*). Remarkably, ABAleon2.15 harboring the PYR1$_{H115A}$ mutation exhibited an apparent affinity for (−)-ABA of ~ 30 µM (*Figure 7G*, *Figure 7—figure supplement 1F,G*), which was 50-fold reduced compared to the (+)-ABA affinity (K'$_d$ ~ 0.6 µ M) (*Figure 7E*, *Figure 7—figure supplement 1C,D*) suggesting, that PYR1$_{H115}$ has an important function in ABA stereospecificity. From these analyses, ABAleon2.13 and ABAleon2.15 could be good candidates for analyzing ABA concentration changes in cell types that have higher basal ABA concentrations.

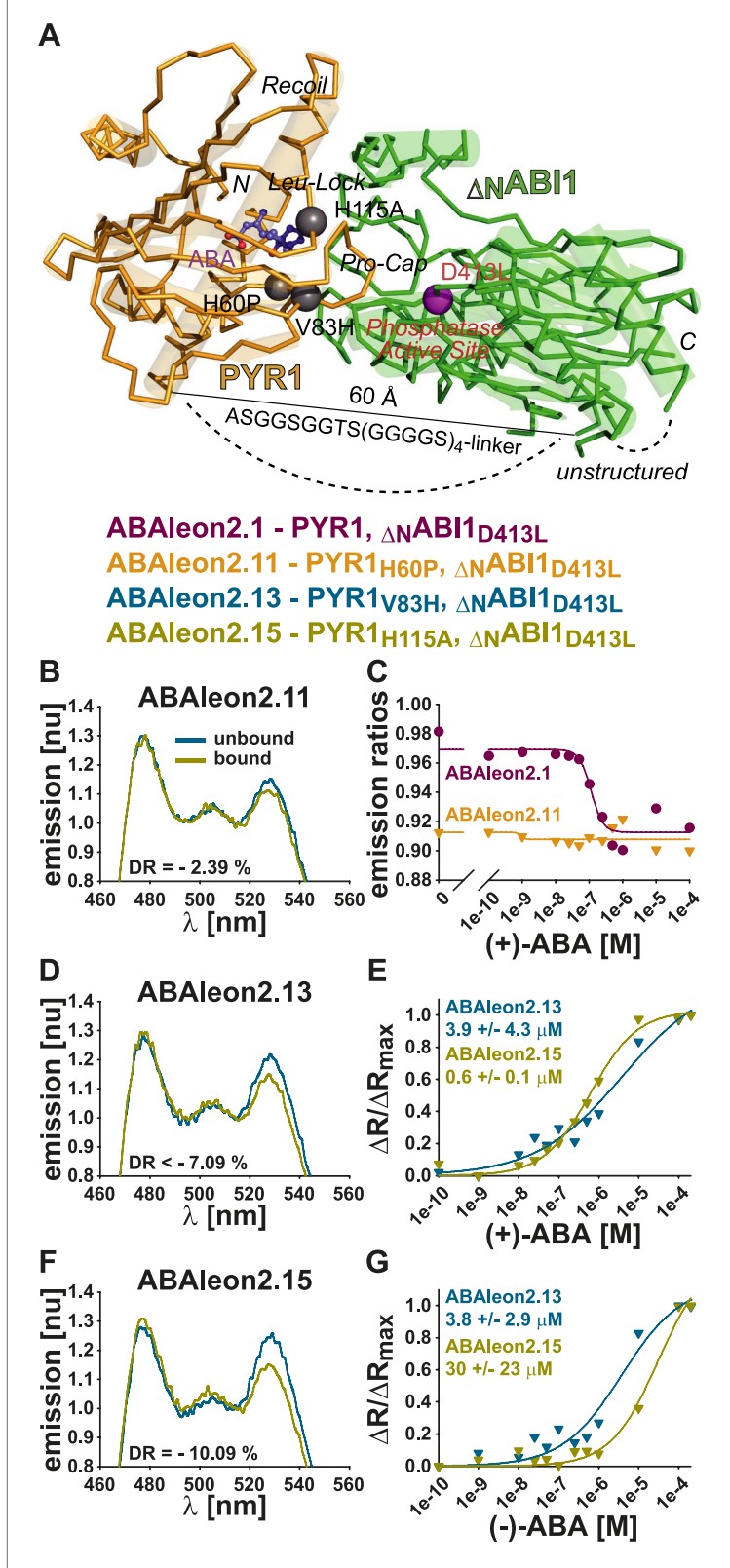

**Figure 7**. *In vitro* analyses of ABAleon2.1 mutants. (**A**) Structural model of the PYR1(gold)-ABA(purple)-ABI1(green) complex, indicating mutations in ABAleon2.1 that were analyzed in (**B**–**G**): H60P monomer-inducing, V83H in Pro-Cap and H115A in Leu-Lock of PYR1 (grey balls) and D413L phosphatase-inactivating in ABI1 (purple ball). Emission spectra

*Figure 7. Continued on next page*

*Figure 7. Continued*

of (**B**) ABAleon2.11, (**D**) ABAleon2.13 and (**F**) ABAleon2.15 in the absence (unbound) and presence of (+)-ABA (bound) with indicated dynamic range (DR). (**B**) ABAleon2.11 exhibited no clear response to ABA. (**C**) (+)-ABA titrations of ABAleon2.11 compared to ABAleon2.1 suggest saturation of ABAleon2.11 in the absence of ABA. (**E** and **G**) ΔR/ΔR$_{max}$ plots of ABAleon titrations with (**E**) naturally occurring (+)-ABA and (**G**) its enantiomer (−)-ABA, which binds more weakly. The respective apparent ABA affinities (K'$_d$) are indicated.

The following figure supplements are available for figure 7:

**Figure supplement 1**. (+)- and (−)-ABA titrations of selected ABAleons.

**Figure supplement 2**. Purification of ABAleons after expression in *E. coli*.

**Table 1.** Biochemical properties of ABAleons

| ABAleon | Mutations/Deletions | R$_{min}$ | R$_{max}$ | DR [%] | K'$_d$ (3 parameter Hill) [nM] | K'$_d$ (4 parameter logistic) [nM] |
|---|---|---|---|---|---|---|
| empty FRET | Δ(PYR1-$_{ΔN}$ABI1) | 2.50 | 2.51 | −0.69 | – | – |
| ABAleon1.1 | – | 0.87 | 0.98 | −14.83 | 266 ± 55 | 332 ± 45 |
| ABAleon2.1 | $_{ΔN}$ABI1$_{D413L}$ | 0.91 | 0.97 | −8.98 | 79 ± 29 | 114 ± 32 |
| ABAleon2.2 | Δ[(GGGGS)$_3$] linker | 0.98 | 1.04 | −8.14 | 121 ± 38 | 156 ± 47 |
| ABAleon2.3 | Δ(GGSGGTS) linker | 0.94 | 0.99 | −7.53 | 72 ± 18 | 84 ± 22 |
| ABAleon2.11 | PYR1$_{H60P}$, $_{ΔN}$ABI1$_{D413L}$ | 0.91 | 0.91 | −2.39 | – | – |
| ABAleon2.12 | PYR1$_{F61L}$, $_{ΔN}$ABI1$_{D413L}$ | 1.05 | 1.12 | −8.29 | 87 ± 20 | 107 ± 22 |
| ABAleon2.13 | PYR1$_{V83H}$, $_{ΔN}$ABI1$_{D413L}$ | 0.91 | 0.97 | −7.09 | 8600 ± 7100 | 2900 ± 1500 |
| ABAleon2.14 | PYR1$_{L87F}$, $_{ΔN}$ABI1$_{D413L}$ | 0.89 | 0.91 | −2.80 | 1200 ± 1200 | – |
| ABAleon2.15 | PYR1$_{H115A}$, $_{ΔN}$ABI1$_{D413L}$ | 0.92 | 1.01 | −10.09 | 488 ± 45 | 510 ± 41 |
| ABAleon2.16 | PYR1$_{E141Q}$, $_{ΔN}$ABI1$_{D413L}$ | 0.97 | 1.02 | −7.30 | 194 ± 46 | 229 ± 48 |
| ABAleon2.17 | $_{ΔN}$ABI1$_{D413L,E142Q}$ | 0.78 | 0.82 | −4.90 | 35 ± 10 | 48 ± 8 |

Biochemical properties of the empty FRET cassette and ABAleons with indicated mutations or deletions compared to the wild type ABAleon1.1. Shown are minimum (R$_{min}$) and maximum emission ratios (R$_{max}$), the dynamic range (DR) calculated as $\frac{R_{min} - R_{max}}{R_{min}} \cdot 100$ and the apparent ABA affinity (K'$_d$) calculated from a three parameter Hill fit or a four parameter logistic fit.

## Low affinity ABAleon2.15 reports ABA uptake in roots

To investigate the utility of low affinity ABAleons, ABAleon2.13, ABAleon2.14 and ABAleon2.15 were transformed into Arabidopsis Col-0 wild type plants. ABAleon2.14 was included in these analyses, as it exhibited an apparent K'$_d$ of ~ 1.2 µM for ABA, however with a reduced dynamic range (*Table 1*). Initial analyses were performed in the root maturation zone of T$_2$ lines and compared with ABAleon2.1 (line 10) at comparable expression levels (*Figure 8*, *Figure 8—figure supplement 1*). Examples of single representative measurements are presented in *Figure 8—figure supplement 1*. All investigated ABAleons responded to externally applied 10 µM ABA with a negative emission ratio change (*Figure 8*, *Figure 8—figure supplement 1*). While ABAleon2.13 and ABAleon2.14 exhibited a relatively low dynamic range *in planta* (*Figure 8B,C,E*, *Figure 8—figure supplement 1E,H*), ABAleon2.15 responded comparable to ABAleon2.1 (line 10) (*Figure 8A,D,E,F*, *Figure 8—figure supplement 1B,K*). Thus, ABAleon2.15 is the best candidate for a low affinity (K'$_d$ ~ 600 nM) ABA-reporter.

## ABAleon2.1 reports endogenous ABA concentration changes in response to low humidity, salt and osmotic stress

It is well established that plants synthesize ABA in response to water stress (*Seo and Koshiba, 2011*). Recent studies reported ABA increases in shoots and roots after 3 h of water stress (*Ikegami et al.,*

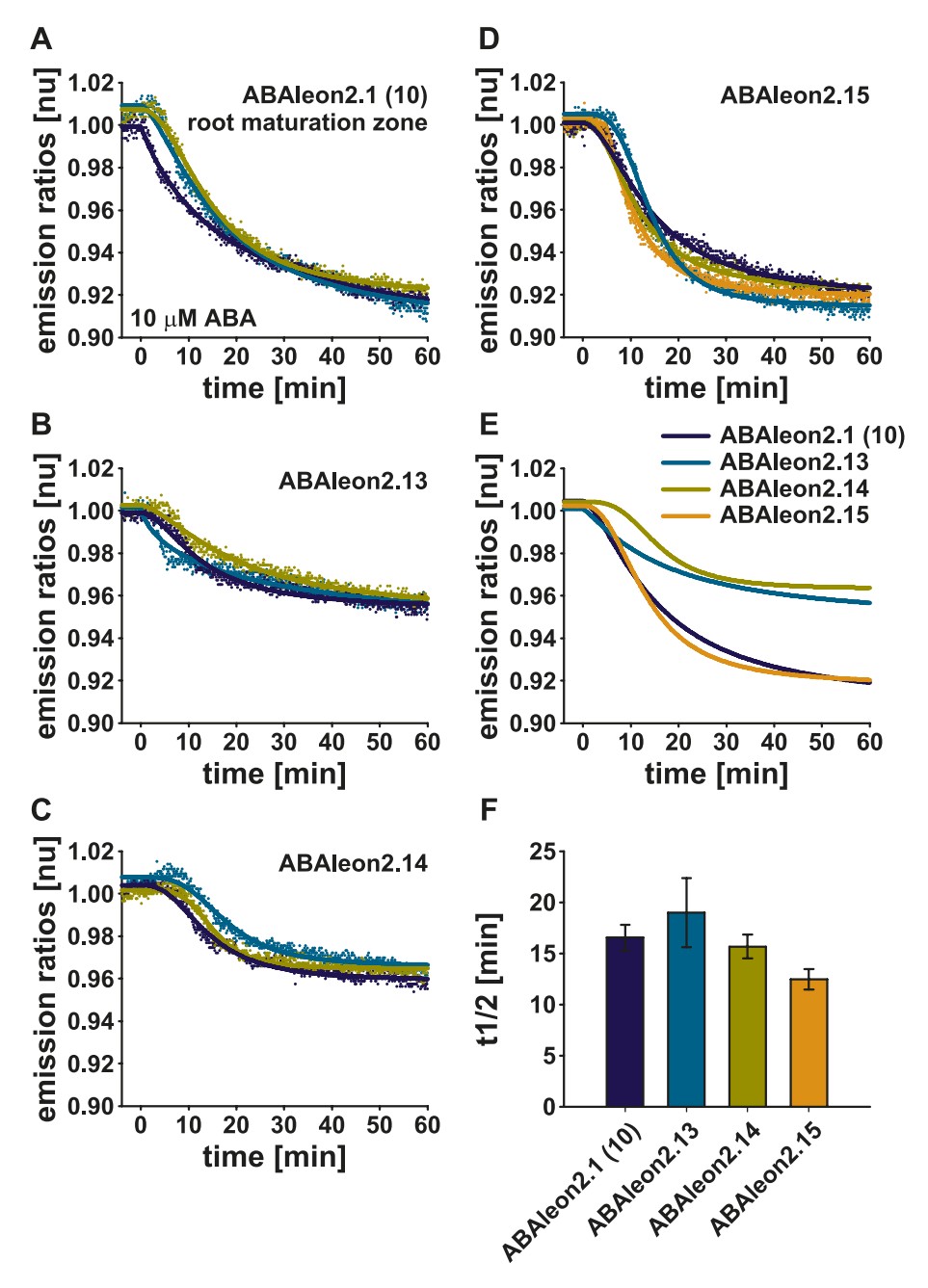

**Figure 8**. ABA-induced ABAleon2.1 (line 10), ABAleon2.13, ABAleon2.14 and ABAleon2.15 responses in the root maturation zone. 10 μM ABA-induced normalized emission ratio changes in the root maturation zone of (**A**, **E** dark blue line) ABAleon2.1 (line 10), (**B**, **E** cyan line) ABAleon2.13, (**C**, **E** yellow line) ABAleon2.14, and (**D**, **E** orange line) ABAleon2.15. (**A–D**) Data points from single measurements fitted by the respective four parameter logistic curve. (**E**) Combined data from three to four experiments in (**A–D**) fitted by the respective four parameter logistic curve. (**F**) $t_{1/2}$ values (means ± SEM, n = 3–4) calculated from the fitted curves in (**A–D**).

The following figure supplements are available for figure 8:

**Figure supplement 1**. ABA-induced ABAleon2.1 (line 10), ABAleon2.13, ABAleon2.14 and ABAleon2.15 responses in the root maturation zone (examples).

*2009*; *Geng et al., 2013*). In addition, older studies reported ABA concentration increases within 15 min in guard cells of *Vicia faba* (*Harris and Outlaw, 1991*).

15 min after a drop in humidity ABAleon2.1 emission ratios decreased ~ 3 % in guard cells and did not change when analyzed at the 30 min time point (*Figure 9A,B*), indicating fast ABA concentration adjustments in response to humidity changes. In 4 h stress treatments of detached leaves 100 mM NaCl and 10 µM ABA induced a 5–6 % ABAleon2.1 emission ratio change in guard cells (*Figure 9C,D*). In contrast, 300 mM sorbitol did not induce any detectable changes (*Figure 9C,D*).

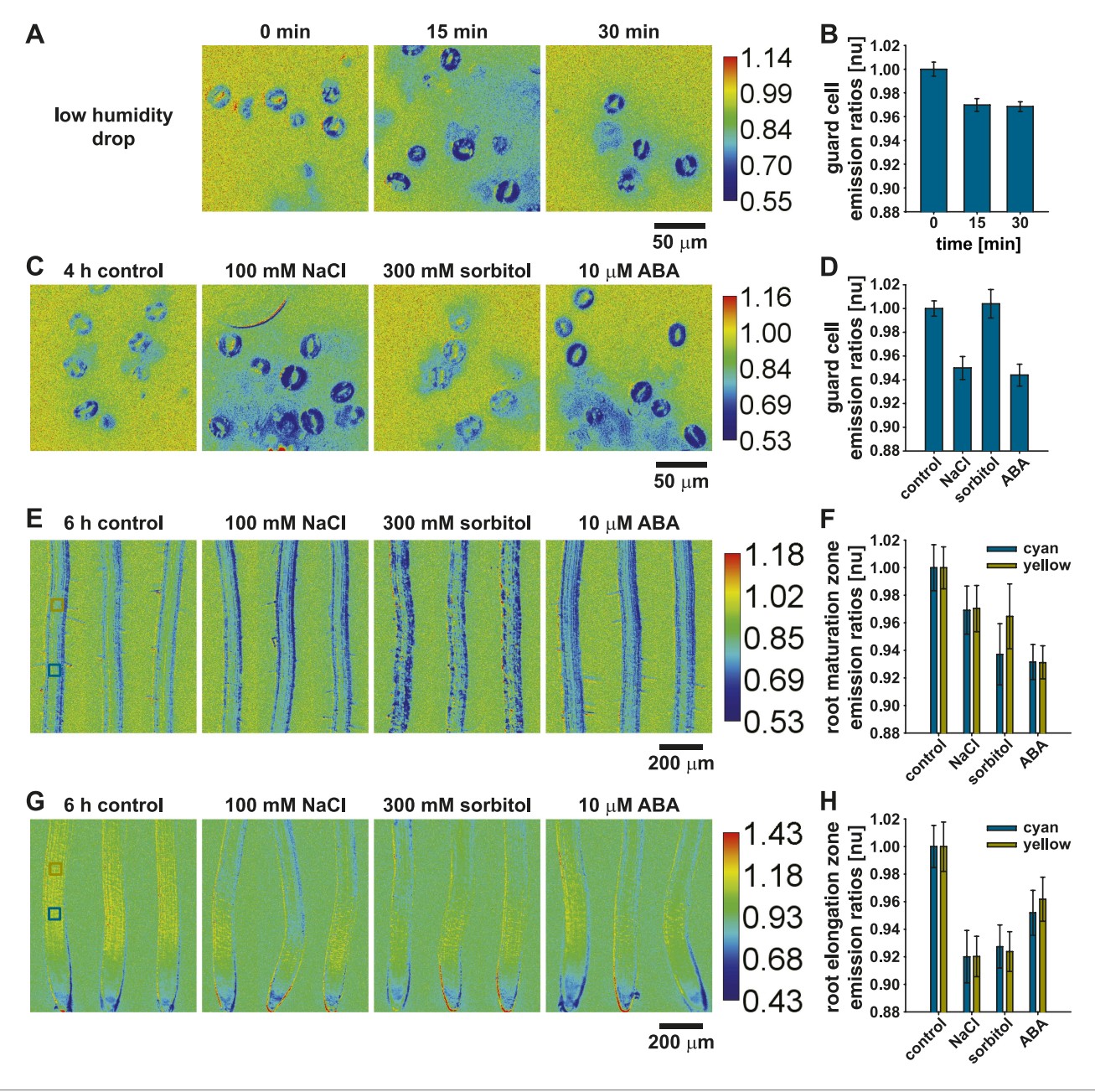

**Figure 9**. ABAleon2.1 reports ABA concentration changes in response to low humidity, salt and osmotic stress. ABAleon2.1 emission ratios in response to (**A** and **B**) low humidity and (**C–H**) 4–6 h treatments with 0.01 % EtOH (control), 100 mM NaCl, 300 mM sorbitol and 10 µM ABA in (**C** and **D**) guard cells, (**E** and **F**) the root maturation- and (**G** and **H**) elongation zone. (**A**, **C**, **E**, **G**) Representative emission ratio images with indicated calibration bars. (**B** and **D**) Normalized emission ratios in guard cells (means ± SEM, n = 3 with ≥ 24 guard cell pairs/n). (**F** and **H**) Normalized emission ratios analyzed from two boxed regions (cyan and yellow) color-coded in the left images of (**E** and **G**) (means ± SEM, n = 8–10 seedlings).

In root tissues two separate regions labeled by cyan and yellow boxes (*Figure 9E,G*, left) were measured after 6 h stress treatments. In these analyses 100 mM NaCl or 300 mM sorbitol induced ABAleon2.1 emission ratio changes of 3–6 % in the root maturation zone compared to 7 % changes in response to 10 μM ABA (*Figure 9F*). In the root elongation zone, responses to 100 mM NaCl and 300 mM sorbitol were 7–8 % while 10 μM ABA induced ABAleon2.1 emission ratio changes of 5–6 % (*Figure 9H*). These analyses demonstrate the utility of ABAleon2.1 to report endogenous ABA concentration changes in response to low humidity, salt- and osmotic stress (*Figure 9*).

## Discussion

### Design of ABAleon reporters

Genetically encoded fluorescent protein-based reporters are powerful tools in cell biology (*Giepmans et al., 2006*; *Okumoto et al., 2012*; *Alford et al., 2013*). Here we report the design, engineering and application of ABAleons, FRET-based reporters that enable the direct analysis of instantaneous ABA concentration changes *in vitro* and *in planta*.

ABAleons were built on the strictly ABA-dependent interaction of PP2Cs with PYR1 and its phylogenetically close relatives (*Park et al., 2009*; *Santiago et al., 2009b*; *Nishimura et al., 2010*), rather than the other members of the PYR/PYL/RCAR family, which have a higher probability of residing in a 'monomeric' state, and which can interact with PP2Cs even in the absence of ABA (*Ma et al., 2009*; *Park et al., 2009*; *Dupeux et al., 2011*; *Hao et al., 2011*). PYR1 and close homologs exhibit lower ABA affinities ($K_d \sim 50–100$ μM) than those of the 'monomeric' homologs ($K_d \sim 1$ μM) (*Ma et al., 2009*; *Miyazono et al., 2009*; *Santiago et al., 2009b*; *Dupeux et al., 2011*). In comparison, the ABA affinity of PYR/PYL/RCARs when bound to PP2Cs ranges between $K_d \sim 20–125$ nM (*Ma et al., 2009*; *Joshi-Saha et al., 2011*). ABAleons exhibit similar ABA affinities to endogenous PYR/PYL/RCARs or PYR/PYL/RCARs in complex with PP2Cs (*Figure 1E,F*, *Figure 7E*, *Figure 7—figure supplement 1*; *Table 1*), consistent with our findings, that ABAleon2.1 is able to detect changes in endogenous ABA concentrations (*Figure 9*).

ABAleons exhibit higher affinity to the naturally occurring (+)-ABA than to (−)-ABA (*Figure 7E,G*, *Figure 7—figure supplement 1B–G*), and could potentially also bind to the synthetic ABA mimics pyrabactin and quinabactin (*Park et al., 2009*; *Okamoto et al., 2013*). One of the ABAleon derivatives, ABAleon2.15, carrying the PYR1$_{H115A}$ mutation, exhibited strongly reduced affinity for (−)-ABA (*Figure 7G*, *Figure 7—figure supplement 1E–G*), suggesting an important role of this amino acid in ABA stereospecificity. These results are consistent with recent findings that the homologous H139 in PYL3 was important for ABA stereospecificity (*Zhang et al., 2013*). ABAleon2.11, carrying the PYR1$_{H60P}$ mutation (*Dupeux et al., 2011*), exhibited spectral characteristics comparable to ABA-bound ABAleon2.1 (*Figure 7C*). This is consistent with the notion, that the PYR1$_{H60P}$ protein may form an alternative interaction interface with PP2Cs, even in the absence of ABA (*Dupeux et al., 2011*).

In Arabidopsis, over-expression of the ABA receptors PYR1, PYL2 and PYL5 induces ABA hypersensitivity (*Figure 5*; *Santiago et al., 2009b*; *Mosquna et al., 2011*), while ectopic expression of the PP2Cs ABI1 and HAB1 decreases ABA sensitivity (*Figure 5*; *Santiago et al., 2009b*; *Nishimura et al., 2010*). Transgenic Arabidopsis plants expressing ABAleon2.1 exhibited a reduced sensitivity to exogenously applied ABA in seed germination, cotyledon expansion and growth assays, but not in stomatal closure assays (*Figure 5*). The degree of reduced ABA sensitivity correlated with increasing ABAleon2.1 expression levels in two independent transgenic lines (*Figure 5*). Because ABAleon2.1 did not exhibit any phosphatase activity *in vitro* (*Figure 1H*), it can be speculated, that ABAleon2.1 might sequester a certain amount of physiologically relevant ABA in certain tissues and cell types due to its high affinity (*Figure 1E,F*, *Figure 7—figure supplement 1C,D*; *Table 1*), thus causing ABA hyposensitivity (*Figure 5*). Future generations of ABAleons and improved imaging sensitivity at lower ABAleon concentrations (*Figure 8*) could enable the reduction of ABAleon side effects.

### Analyses of ABA concentration changes and long-distance ABA transport in Arabidopsis

By definition, a hormone transmits a signal from the site of hormone synthesis to its place of action. Although long-distance ABA transport has been studied for many years (*Sauter et al., 2001*; *Wilkinson and Davies, 2002*; *Seo and Koshiba, 2011*; *Boursiac et al., 2013* and references therein), no method for the direct detection of ABA concentration changes and ABA transport rates *in planta* has been

available. The detailed characterizations of ABAleons demonstrate the utility of ABAleon2.1 ($K'_d$ ~ 100 nM) and ABAleon2.15 ($K'_d$ ~ 600 nM), which exhibit a sufficient ABA-specificity and dynamic range (9–10 %) upon ABA binding to monitor instantaneous ABA-induced or environmentally-triggered ABA concentration changes.

ABAleon2.1 and ABAleon2.15 enabled measurements of rapid ABA-induced changes in ABA concentrations in various tissues (*Figure 3*, *Figure 8*), ABA uptake into whole seedlings (*Figure 2A,B*), directional ABA transport from the hypocotyl base towards the shoot (*Figure 3E,F*; *Video 1*) and from the hypocotyl or shoot to the root (*Figure 6B,C,E-H*). Under the imposed conditions the speed of ABA transport within the hypocotyl was ~ 16 µm/min. Furthermore, ABA transport from the root maturation zone to the shoot could not be detected within three hours (*Figure 6D,I,J*). Because the experimental setup (*Figure 6*), in which both shoot and root were perfused with buffer, would compromise the transpirational stream, the present data do not exclude concomitant ABA transport from roots to shoots, as has been found in other plant species (*Wilkinson and Davies, 2002*).

In response to water stress, Arabidopsis plants synthesize ABA in the shoot, which has been reported to be transported to the root (*Ikegami et al., 2009*). ABA accumulation in roots and leaves was detected 2.5–3 h after stress initiation (*Ikegami et al., 2009*; *Geng et al., 2013*). Older studies, using manually dissected guard cells, measured ABA concentration increases in guard cells 15 min after passive dehydration of leaves (*Harris and Outlaw, 1991*). ABAleon2.1 enabled the rapid detection of ABA concentration changes in guard cells in response to a humidity drop (*Figure 9A,B*) and the visualization of long term ABA accumulation in response to salt in guard cells (*Figure 9C,D*) and in response to salt- and osmotic stress in roots (*Figure 9E–H*).

Surprisingly, ABA-induced ABAleon2.1 ratio changes in the root maturation zone were accelerated in the *pyr1-1/pyl1-1/pyl2-1/pyl4-1* mutant (*Figure 4*). Mutants defective in ABA signaling may compensate by up-regulating ABA levels, as reported previously (*Nakashima et al., 2009*) and could also up-regulate ABA transport activity (*Figure 4C,D*). Alternatively, knock out of ABA receptors may also affect ABA buffering capacity (*Figure 4C,D*). ABA reporter analyses of ABA uptake (*Figure 4*) or long-distance ABA transport (*Figure 3D–F*, *Figure 6*) could be utilized for the characterization or identification of ABA transporters and their regulation mechanisms *in planta* or in heterologous systems (*Jones et al., 2014*).

ABA induces cytoplasmic alkalinization of guard cells (*Blatt and Armstrong, 1993*; *Islam et al., 2010*). In guard cells of *Vicia faba*, cytoplasmic pH was found to be 7.67 and increased 0.27 units upon ABA treatment (*Blatt and Armstrong, 1993*). Cytoplasmic pH in roots was 7.3 and could increase to 7.7 (*Bibikova et al., 1998*). In these pH ranges ABA-induced ABAleon1.1 emission ratio changes were stable *in vitro* (*Figure 1—figure supplement 1B*).

*In vivo* ABA concentrations in cellular compartments of specific plant cells and tissues are currently unknown. Overall ABA levels range from 30–50 ng/g dry-weight in non-stressed plants (*Forcat et al., 2008*; *Geng et al., 2013*), which can increase up to 30-fold in response to limited water conditions (*Harris et al., 1988*; *Harris and Outlaw 1991*; *Christmann et al., 2007*; *Ikegami et al., 2009*; *Geng et al., 2013*). ABAleon2.1 and ABAleon2.15 exhibit a sufficient dynamic range for *in vitro* calibrations (*Figure 1E,F*, *Figure 7E*, *Figure 7—figure supplement 1B–D*) that permit approximations of cellular ABA concentrations, for example in the range of ≤ 25 nM in the root elongation zone. In *Vicia faba* guard cells ABA concentrations were ~ 0.7 fg/cell pair in unstressed and ~ 17.7 fg/cell pair in stressed guard cells (*Harris et al., 1988*). ABA concentrations in stressed guard cells were estimated to be in the range of ~ 15 µM (*Harris et al., 1988*; *Harris and Outlaw 1991*). Extrapolating from these values, unstressed guard cell ABA concentration would be ~ 500 nM. Such approximations would be consistent with the partial saturation and reduced response of ABAleon2.1 in guard cells (*Figure 3A–C*) and with strong expression of the ABA-induced reporter pRAB18-GFP (*Figure 2—figure supplement 1*).

Our results demonstrate that ABAleons affect ABA signaling to certain extent, but can analyze changes in ABA concentrations in diverse tissues and cell types and measure ABA transport *in vivo*. ABAleons will thus allow hitherto challenging investigations of ABA synthesis and transport *in planta*, in response to changes in environmental conditions or treatment with synthetic compounds designed to improve plant survival and crop yields under adverse climate conditions. Further, in combination with other genetically encoded reporters, ABAleons can be used to decipher the cross talk between ABA and other signaling molecules. During the course of our ABAleon research we found, that Jones et al. developed ABACUS-type ABA reporters, however with biochemical properties complementary

to ABAleons (*Jones et al., 2014*). Thus, ABAleons and ABACUS could be utilized to study novel aspects of ABA signaling in intact plants.

## Materials and methods

### Construction of ABAleons

Fluorescent protein coding sequences were re-amplified from the pF40 plasmid (*Piljić et al., 2011*) and ligated into *Xba I/Apa I* (mTurquoise) or *Xma I/Sac I* sites (cpVenus173) of a modified pUC19 plasmid (*Walter et al., 2004*; *Waadt et al., 2008*) resulting in the pUC-F3 and pUC-F3_II empty FRET-cassettes with the latter containing a *Nde I*-site downstream of the *Xba I*-site and a StrepII-tag fusion of cpVenus173 at its free end. PYR1-GGSGG and (GGGGS)$_{4}$-$_{\Delta N}$ABI1 and mutants were cloned and inserted between *Apa I/Spe I* and *Spe I/Xma I* sites of pUC-F3 and pUC-F3_II to obtain pUC-ABAleon and pUC-ABAleon_II, respectively. *Escherichia coli* expression vectors were obtained by sub-cloning ABAleon_II versions *Nde I/Sac I* into pET-24b(+) (Novagen, Darmstadt, Germany). For expression in plants, the pUBQ10 promoter (AT4G05310; *Norris et al., 1993*; *Krebs et al., 2012*), inserted between *Hind III/Spe I* sites of a modified pUC19 plasmid (kindly provided by Jörg Kudla, University of Münster), was mutated to remove a *Sac I* site within the pUBQ10. In addition, the HSP18.2 terminator (T) (AT5G59720; *Nagaya et al., 2010*) was inserted between the *Sac I/Eco RI* sites of pBluescript II (Stratagene, La Jolla, CA) and the *Hind III* site within the HSP18.2T was deleted resulting in pKS-HSP18.2T$_{\Delta Hind III}$. Both pUBQ10$_{\Delta Sac I}$ and HSP18.2T$_{\Delta Hind III}$ were sub-cloned into plant compatible vectors pGPTVII.Bar, which confers glufosinate (BASTA) resistance, and pGPTVII. Hyg, which confers hygromycin resistance (*Walter et al., 2004*), resulting in the barII-UT and hygII-UT plasmids. Finally, ABAleon2.1, ABAleon2.1 mutants and the empty FRET-cassette were sub-cloned from pUC-ABAleon2.1, pUC-ABAleon2.1x_II and pUC-F3 plasmids into the barII-UT and hygII-UT plasmids to obtain the barII-UT-ABAleon2.1, barII-UTF3 and hygII-UT-ABAleon2.1 and mutant plasmids for expression in plants. More detailed information about oligo-nucleotides and plasmids used and generated in this work is provided in *Supplementary file 1A* and *Supplementary file 1B*.

### Protein expression and purification

pET-F3_II (empty FRET) and pET-ABAleon_II versions in BL21-CodonPlus (DE3)-RIL cells (Stratagene) were grown at 37 °C in 2 L Luria Broth (LB) medium containing 25 μg/mL Kanamycin. At an optical density at 600 nm (OD$_{600}$) of 0.5, cells were induced with 0.5 mM Isopropyl β-D-1-thiogalactopyranoside (IPTG) and shaken for additional 4-6 h at 24 °C. Cells were collected by centrifugation (15 min 5.000×*g* and 4 °C) and stored at −80 °C. Proteins were extracted by sonification after 60 min incubation in 20 mL extraction buffer (30 mM Tris–HCl [pH 7.4], 250 mM NaCl, 1 mM Ethylenediaminetetraacetic acid [EDTA], 1 mM Phenylmethylsulfonyl fluoride [PMSF], 1x protease inhibitor [Roche, USA] and 1 mg/mL Lysozym). Cell debris was removed by centrifugation (2 × 30 min, 20.000×*g* and 4 °C) and by filtration through 0.45 μm syringe filters.

Protein extracts were mixed with 2.5 ml 50 % Strep-Tactin Macroprep resin (IBA, Göttingen, Germany) pre-equilibrated in wash buffer I (30 mM Tris–HCl [pH 7.4], 250 mM NaCl, 1 mM EDTA) and protein/resin mix was incubated for 1 h at 4 °C while shaking in a 50 ml tube. The suspension was run twice through a 20 mL gravity column (BioRad, Hercules, CA) followed by two washes of the remaining protein/resin mix with 10 column volumes (CV) of wash buffer I and one wash with 10 CV of wash buffer II (30 mM Tris–HCl [pH 7.4], 250 mM NaCl, 10 mM MgCl$_2$ and 1 mM MnCl$_2$). Proteins were eluted 3x in 1 CV wash buffer II supplemented with 2.5 mM Desthiobiotin (Sigma, USA) and concentrated to ~ 1 mL volume by centrifugation at 3.000×*g* and 4 °C using Amicon Ultra-4 30K or 10K centrifugal filters (Millipore, Billerica, MA). Purified proteins were run through a Superdex 200 HiLoad 16/60 column (GE Healthcare) in wash buffer II using an ÄKTA purifier fast protein liquid chromatography (FPLC) system (GE Healthcare) with 0.8 MPa column pressure limit, 1 mL/min flow rate and 2 mL fraction size volume. Fractions exhibiting 280 nm, 445 nm and 516 nm absorbance were analyzed by SDS-PAGE and Instant Blue (Cole–Parmer, USA) protein staining and selected for further concentration using Amicon Ultra-4 centrifugal filters. Protein aliquots were flash frozen in liquid N$_2$ and stored at −80 °C. Protein purity was analyzed by SDS-PAGE, immuno-blotting using anti-GFP antibody (Life Technologies, Darmstadt, Germany) and PageBlue staining (Thermo Scientific, Rockford, IL; *Waadt et al., 2014*). Results of F3 empty FRET and ABAleon purifications are provided in *Figure 7—figure supplement 2*.

The coding sequence of $_{\Delta N}$ABI1 corresponding to amino acid residues 125–429 was inserted via *Nde I*/*Bam HI* into pET28a (Novagen; *Supplementary file 1A*, *Supplementary file 1B*) and transformed into *E. coli* BL21 (DE3). *E. coli* were grown at 37 °C, and protein expression was induced by 1 mM IPTG at $OD_{600}$ of 0.6–0.8. After overnight incubation at 25 °C, cells were harvested by centrifugation (15 min 5.000×*g* and 4 °C) and re-suspended in 50 mM Tris–HCl (pH 8.0) and 500 mM NaCl. Cells were sonicated on ice and lysates were obtained after centrifugation at 12,000×*g* for 1 h 6xHis-$_{\Delta N}$ABI1 extracts were applied to a Ni-NTA column (Qiagen, Hilden, Germany) and washed with five bed volumes of 50 mM Tris–HCl (pH 8.0), 500 mM NaCl and 10 mM imidazole. Bound proteins were eluted in 50 mM Tris–HCl (pH 8.0), 500 mM NaCl and 300 mM imidazole. 6xHis-$_{\Delta N}$ABI1 was further purified using Sephacryl S-200 (GE Healthcare) in 50 mM Tris–HCl (pH 8.0) and 150 mM NaCl. 6xHis-PYR1 (*Nishimura et al., 2009*) and 6xHis-$_{\Delta N}$ABI1 proteins were re-buffered and concentrated into wash buffer II using Amicon Ultra-4 centrifugal filters. Protein purity and concentrations were analyzed by SDS-PAGE and PageBlue staining and quantified according to a 0–2000 ng BSA (NEB, Ipswich, MA) standard calibration.

### *In vitro* analyses of ABAleons

ABA titration experiments were conducted in a TECAN Infinite M1000 PRO (TECAN, Männedorf, Switzerland) using 1 µM ABAleon protein in wash buffer II with 0.1 % EtOH or DMSO as solvent for (+)-ABA (TCI, Portland, OR), generally used in assays unless otherwise stated, or (−)-ABA (Sigma). Protein samples were excited with 440 ± 5 nm and emission 450–700 nm was measured in 1 nm steps with 5 nm bandwidth and 10 flashes of 20 µs and 400 Hz. Gain settings to obtain optimal emission spectra were calculated by the TECAN software from unbound ABAleon emission. Emission bands of mTurquoise (470–490 nm) and cpVenus173 (518–538 nm) were used to calculate cpVenus173/mTurquoise emission ratios. Apparent ABA affinities ($K'_d$) were calculated from emission ratio plots by fitting a four parameter logistic curve $R = R_{min} + \dfrac{R_{max} - R_{min}}{1 + (\frac{[ABA]}{K'd})^n}$ or from ΔR/ Δ $R_{max}$ plots by fitting a three parameter sigmoidal Hill equation $\dfrac{\Delta R}{\Delta R_{max}} = \dfrac{R_{max} \cdot [ABA]^n}{K_d^{'n} \cdot [ABA]^n}$ (*Palmer et al., 2006*) using the SigmaPlot 10.0 version (Systat, San Jose, CA). Dynamic ranges were calculated from experimentally determined values as $\dfrac{R_{min} - R_{max}}{R_{min}} \cdot 100$.

Absorbance spectra (275–700 nm, slit 0.2 nm) of ABAleons were analyzed with a UV-VIS-Spectrophotometer (UV-2700) (Shimadzu, Columbia, MD). Absorbance at 434 nm (mTurquoise) was used to calculate concentrations of the empty FRET and ABAleon proteins (*Goedhart et al., 2010*) and the ratio of $Abs_{515}$ (cpVenus173) and $Abs_{434}$ was used to estimate protein purity.

pH titrations were performed by addition of 1 µL concentrated ABAleon1.1 protein (final concentration 200 nM) to 100 µL wash buffer II adjusted to a pH range between pH 5.0–8.2 with 1 M HCl and MES powder or with 2 M NaOH. After recordings of ABA-free ABAleon1.1 emission spectra, using the TECAN Infinite M1000 PRO as mentioned above, 1 µL of 1 mM ABA (final concentration 10 µM ABA) was added and emission spectra were recorded using identical settings. Experiments were performed in duplicate and fitted by a four parameter Hill equation $R = R_0 + (\dfrac{a \cdot x^n}{c^n + x^n})$ in SigmaPlot 10.0 (Systat, San Jose, CA). Ratio change was calculated by subtraction of the ABA-free from the ABA-bound equation values.

ABA-induced ABAleon2.1 kinetics were analyzed using 2.77 µM ABAleon2.1 in a Berthold Mithras LB 940 (Berthold Technologies, Bad Wildbad, Germany) with the following settings: Lamp energy 5000, counting time 0.05 s, excitation 440 ± 10 nm, emission 470 ± 10 nm (mTurquoise) and 530 ± 20 nm (cpVenus173) measured in cycles of 6.12 s. At cycle 25, 50 µL 3 µM ABA in wash buffer II and 0.3 % EtOH was applied with low injector speed to result in the final 1 µM ABA in 0.1 % EtOH.

Phosphatase assays were performed using the serine/threonine phosphatase assay system (Promega, Madison, WI). In brief, 50 µL reactions containing wash buffer II, 10-50 pmol protein, 5000 pmol Ser/Thr phosphopeptide ± 5 µM ABA with 0.005 % EtOH as solvent were incubated for 10–30 min at room temperature. Reactions were stopped by addition of 50 µL of the supplied molybdate dye/additive mixture and phosphate release was measured according to a standard curve with a Berthold Mithras LB 940 plate reader (Absorbance 600 ± 10 nm, lamp energy 50,000 and counting time 2 s).

## Structural modeling of ABAleon

Three-dimensional coordinates of major components of ABAleon were built with known crystal structures of mTurquoise (pdb: 2YE0, *Goedhart et al., 2012*), PYL1-ABA-ABI1 (pdb: 3JRQ, *Miyazono et al., 2009*) and Venus (pdb: 1MYW, *Rekas et al., 2002*). Each component of ABAleon was manually assembled using COOT (*Emsley et al., 2010*). In the assembly, PYL1 was replaced by PYR1 (pdb: 3K3K, *Nishimura et al., 2009*) by tracing the Cα backbone. The unstructured C-terminus of mTurquoise was placed in distance corresponding to the PG linker and the PYR1 N-terminus. Venus was placed between ABI1 and mTurquoise with the N-terminus of Venus facing towards the C-terminus of ABI1. All structural figures were drawn with PyMOL (The PyMOL Molecular Graphics System, Version 1.5.0.4 Schrödinger, LLC.).

## Plant culture and transgenic Arabidopsis lines

Seeds were sterilized in 70 % EtOH and 0.04 % sodium dodecyl sulfate (SDS), washed three times in 100 % EtOH and sown on 0.5 Murashige and Skoog (MS) media (Sigma) adjusted to pH 5.8 with 1 M KOH and supplemented with 0.8 % phytoagar. After at least 4 days of stratification in the dark at 4 °C plants were cultivated in a growth room (16 h day/8 h night cycle, 25 °C, 50–100 $\mu Em^{-2}s^{-1}$ and 26 % relative humidity) or in a CMP4030 plant growth chamber (16 h day/8 h night cycle, 22 °C, 50 $\mu Em^{-2}s^{-1}$ and 25 % relative humidity; Conviron, Winnipeg, Manitoba). 6-day-old seedlings were transferred to soil and grown either in the growth room or in a CMP3244 plant growth chamber (16 h day, 22 °C/8 h night, 18 °C cycle, 50–100 $\mu Em^{-2}s^{-1}$ and 30–50 % relative humidity; Conviron).

barII-UTF3 empty FRET, barII-UT-ABAleon2.1, hygII-UT-ABAleon2.13, hygII-UT-ABAleon2.14 and hygII-UT-ABAleon2.15 were transformed into Arabidopsis Columbia 0 accession and hygII-UT-ABAleon2.1 was transformed into *pyl4ple* [*pyr1-1* (Q169 stop)/*pyl1-1* (SALK_054640)/*pyl2-1* (GT2864)/*pyl4-1* (SAIL_517_C08)] (*Park et al., 2009*; *Nishimura et al., 2010*) by the floral dip method (*Clough and Bent, 1998*). Transformants were selected on 0.5 MS media supplemented with either 10 µg/mL glufosinate or 25 µg/mL hygromycin and further cultivated in soil in a CMP3244 plant growth chamber. Positive transformants were further selected by fluorescence intensity at a confocal microscope (see below). A list of transgenic Arabidopsis lines generated and used in this work is provided in *Supplementary file 1C*.

## ABA sensitivity analyses

For ABA seed germination assays seeds were sown on 0.5 MS agar media supplemented with 0.08 % EtOH as solvent control or 0.8 µM (+)-ABA (TCI, Portland, OR). After stratification, plants were grown in the growth room. Germinated seeds and seedlings with expanded green cotyledons were counted for a time period of 7 days with blinded genotypes. Analyses represent mean values ± SEM of four technical replicates normalized to the seed count (48–50 seeds) of each experiment.

For growth assays, 4-day-old seedlings grown on 0.5 MS agar media were transferred to 0.5 MS agar media supplemented with 0.1 % EtOH as solvent control or 10 µM (+)-ABA and grown vertically in the growth room. Images were acquired 5 days after seedling transfer. Fresh weight was measured from pools of seven seedlings/experiment (means ± SEM, n = 5) normalized to the 0 µM ABA control conditions.

ABA-induced stomatal closure analyses were performed with 20-23-day-old plants grown vertically on 0.5 MS agar media in the CMP4030 plant growth chamber. Six detached leaves were floated in assay buffer (5 mM KCl, 50 µM CaCl₂ and 10 mM MES-Tris pH 5.6) at 22 °C and 100 µE m$^{-2}$s$^{-1}$ for 2 h. Subsequently, (+)-ABA or EtOH as solvent control was added to a final concentration of 10 µM (+)-ABA in 0.1 % EtOH followed by additional 2 h incubation. Leaves were blended 4x ~10 s in ~50 ml deionized water and leaf epidermal tissue was collected through a 100 µm nylon mesh (Millipore, Billerica, MA) and mounted on a microscope slide for imaging. Images were acquired using an inverted light microscope (Nikon Eclipse TS100) equipped with a 40x/0.65 ∞/0.17 WD. 0.57 objective and connected to the Scion camera and Scion VisiCapture Application Version1.3 (Scion Corporation, Frederick, MD). Experiments were performed with blinded genotypes and treatments and ≥ 24 stomatal apertures were measured per experiment using Fiji (*Schindelin et al., 2012*). Data represent mean stomatal apertures ± SEM of three experiments normalized to the solvent control.

## Sample preparation and microscopic analyses

For guard cell imaging, 4-week-old detached leaves without mid vein were glued with the abaxial side on a cover glass using medical adhesive (Hollister, Libertyville, IL) and upper cell layers were dissected

away with an industrial razor blade. Epidermal strips were incubated in assay buffer (5 mM KCl, 50 µM CaCl₂, 10 mM MES-Tris, pH 5.6) and 0.01 % EtOH, as solvent control for ABA, for 1 h. Glass cover slips were mounted on a microscope slide with a central hole (Ø = 13 mm) using vacuum grease silicone (Beckman, Pasadena, CA) and analyzed in 200 µL of buffer mentioned above. For ABA application, epidermal strips were perfused by washing (pipetting) four to five times with assay buffer supplemented with 10 µM ABA in 0.01 % EtOH. Low humidity drop experiments were conducted on 19-27-day-old seedlings grown vertically on 0.5 MS agar media in the CMP4030 plant growth chamber. Low humidity was induced by opening the lid of the 0.5 MS agar plates. At time points 0, 15 and 30 min after plate opening two seedlings were blended 4x ~10 s in ~50 mL deionized water and leaf epidermal tissue was collected through a 100 µm nylon mesh (Millipore, Billerica, MA) and mounted on a microscope slide for imaging. Experiments were performed in triplicates and ≥ 27 guard cell pairs were analyzed per experiment. Long term stress treatments were conducted on detached leaves of 20-24-day-old plants grown vertically on 0.5 MS agar media in the CMP4030 plant growth chamber. Four leaves were pre-incubated for 1–4 h in assay buffer at 22 °C and 100 µEm⁻²s⁻¹ and treatments were performed by addition of assay buffer supplemented with 10-fold concentrated EtOH (as solvent control for ABA), ABA, NaCl or sorbitol to obtain final concentrations of 0.01 % EtOH, 10 µM ABA, 100 mM NaCl or 300 mM sorbitol. 4 h after the treatments leaves were blended (see above) and leaf epidermal tissue was collected for imaging. Experiments were performed in triplicates with blinded treatments and epidermal fractions used for ABAleon2.1 emission ratio imaging were selected using the bright field channel. 24–40 guard cell pairs were analyzed per experiment.

For seedling imaging, 4-day-old seedlings were transferred to glass bottom dishes (MatTek, Ashland, MA) supplemented with 200 µL 0.25 MS, 10 mM MES-Tris (pH 5.6) and 0.7 % low melting point agarose (Promega) and grown vertically for an additional day in the CMP4030 plant growth chamber. Before microscopic analyses, 90 µL assay buffer was added. Treatments were conducted by pipetting 10 µL ABA solution or the respective amount of EtOH as solvent control diluted in assay buffer to reach a final concentration of 10–50 µM ABA and 0.01–0.05 % EtOH. To apply ABA to defined tissues, transparent modeling clay was used to divide each glass bottom dish into two isolated chambers before application of the growth media. Seedlings were placed on top of the growth media and modeling clay, such that either the hypocotyl base and root differentiation zone (shoot to root transport) or the root maturation zone (hypocotyl to root and root to hypocotyl transport) laid on the dry modeling clay. Long term treatments were performed on 5-day-old seedlings, which were transferred to glass bottom dishes (MatTek, Ashland, MA) supplemented with 200 µL 0.25 MS, 10 mM MES-Tris (pH 5.6) and 0.7 % low melting point agarose with addition of 0.01 % EtOH (as solvent control for ABA), 10 µM ABA, 100 mM NaCl or 300 mM sorbitol. 6 h after transfer 100 µL assay buffer supplemented with 0.01 % EtOH, 10 µM ABA, 100 mM NaCl or 300 mM sorbitol was added before the root maturation- and elongation zone were imaged. Regions of the root maturation zone with similar distance from the root tip were selected in the bright field channel before ratio images were acquired. Treatments were performed blinded and 8–10 seedlings were analyzed per treatment.

pRAB18-GFP plants (*Kim et al., 2011*) were grown in soil in the CMP3244 plant growth chamber. To ensure high relative humidity (RH) conditions (70%) plants were kept under a plastic cover and sprayed with water twice a day. 2 days before the analyses, plants were removed from the growth chamber, placed in 25 % RH conditions and withheld from water supply. For the ABA treatment, detached leaves were floated for 4 h in 50 µM ABA prior to microscopic analyses.

Expression analyses based on cpVenus173, YFP or GFP emission were performed using an Eclipse TE2000-U microscope equipped with Plan 20x/0.40 ∞/0.17 WD 1.3 and Plan Apo 60x/1.20 WI ∞/0.15–0.18 WD 0.22 objectives (Nikon), a CascadeII 512 camera (Photometrics), a MFC2000 z-motor (Applied Scientific Instruments, Eugene, OR), a QLC-100 spinning disc (VisiTech international, Sunderland, UK), a CL-2000 Diode pumped crystal laser (LaserPhysics Inc., West Jordan, UT), a LS 300 Kr/Ar laser (Dynamic Laser, Boston, MA) and guided by Metamorph software version 7.7.7.0 (Molecular Devices). Images were analyzed, processed and calibrated in Fiji (*Schindelin et al., 2012*).

ABAleon ratio-imaging was conducted according to *Allen et al. (1999)* using an Eclipse TE300 microscope equipped with a Plan Fluor 10x/0.30 DIC L ∞/0.17 WD 16.0 for seedlings or a Plan Fluor 40x/1.30 Oil objective DIC H ∞/0.17 WD 0.2 for guard cells (Nikon, Tokyo, Japan), a Cool SNAP HQ camera (Photometrics, Tucson, AZ), a Mac 2002 System automatic controler, a CAMELEON filter set 71007A (D440/20, D485/40, D535/30; Chroma, Bellows Falls, VT) and guided by the MetaFluor software version 7.0r3 (Molecular Devices, Sunnyvale, CA). Images were acquired in intervals of 6 s using

200-250 ms exposure, Binning 4, Gain 2x (4x) and 20 MHz transfer speed. Images were analyzed and processed using Fiji (*Schindelin et al., 2012*). Analyses of seedlings were standardized as treatments were performed 4 min after the experiments started and regions used for emission ratio analyses had identical areas and distances from each other. ABA response curves in the root maturation zone and the hypocotyl were analyzed by fitting $R = R_{min} + \dfrac{R_{max} - R_{min}}{1 + (\frac{t}{t_{1/2}})^n}$ , using SigmaPlot 10.0 (Systat, San Jose, CA), to either data of single measurements or combined datasets.

## Acknowledgements

We thank Ashley J Pratt for help with *in vitro* analyses, members of the Schroeder lab, especially Hans-Henning Kunz, Shintaro Munemasa, Felix Hauser, Jiyoung Park and Benjamin Brandt for providing seeds, plasmids and for advice with ratiometric imaging. We thank Christian Waadt for assembling *Video 1*, Roger Y Tsien (UC San Diego) for providing generous access to equipment and for discussion, and Jörg Kudla (University of Münster) and Carsten Schultz (EMBL, Heidelberg) for providing plasmids. Research analyzing salt stress responses was supported by a grand from the Division of Chemical Sciences, Geosciences, and Biosciences, Office of Basic Energy Sciences of the U.S. Department of Energy (DE-FG02-03ER15449) to JIS.

## Additional information

### Funding

| Funder | Grant reference number | Author |
|---|---|---|
| National Institutes of Health | GM060396-ES010337 | Julian I Schroeder |
| National Science Foundation | MCB-0918220 | Julian I Schroeder |
| Division of Chemical Sciences, Geosciences, and Biosciences, Office of Basic Energy Sciences of the U.S. Department of Energy | DE-FG02-03ER15449 | Julian I Schroeder |
| Alexander von Humboldt-Foundation | Feodor Lynen Research Fellowship | Rainer Waadt |
| National Science Foundation | MCB-1330856 | Elizabeth D Getzoff |

The funders had no role in study design, data collection and interpretation, or the decision to submit the work for publication.

### Author contributions

RW, Conception and design, Acquisition of data, Analysis and interpretation of data, Drafting or revising the article; KH, Acquisition of data, Analysis and interpretation of data, Drafting or revising the article, Contributed unpublished essential data or reagents; NN, Conception and design, Acquisition of data, Analysis and interpretation of data, Contributed unpublished essential data or reagents; CH, SRA, Acquisition of data, Analysis and interpretation of data; EDG, Analysis and interpretation of data, Drafting or revising the article; JIS, Conception and design, Analysis and interpretation of data, Drafting or revising the article

## Additional files

### Supplementary files

• Supplementary file 1. (**A**) Oligo-nucleotides used in this work. List of oligo-nucleotides with indicated Arabidopsis GeneBank ID (AGI) number of the gene or construct, the oligo-nucleotide name and 5'-3'-sequence, the restriction sites included in the oligo-nucleotide and the description for what it was used for. In the oligo-nucleotide sequences the restriction sites are indicated by italic letters and mutations or non gene-coding nucleotides are indicated by lower case letters. (**B**) Plasmids and constructs used and generated in this work. List of plasmids and constructs with indicated Arabidopsis GeneBank ID (AGI) number of the inserted gene or construct, the promoter

included in the plasmid, the clone name, the restriction sites, which were used for cloning, the vector backbone of the clone with incorporated selection markers for bacteria and plants and the clone information. The clone information includes additional information about restriction sites, promoters, inserts and point mutations. Amp, ampicillin; Bar, BASTA; Hyg, hygromycin; Kan, kanamycin; XFP, fluorescent protein. (C) Transgenic Arabidopsis lines used and generated in this work. List of transgenic *Arabidopsis thaliana* plants with indicated ecotype, Arabidopsis GeneBank ID (AGI) of the mutated genes, gene names, mutant names and IDs, the name of the transgenic lines, the construct used for transformation, plant selection marker and description. Bar, BASTA; Hyg, hygromycin; Kan, kanamycin.

## Major datasets

The following previously published datasets were used:

| Author(s) | Year | Dataset title | Dataset ID and/or URL | Database, license, and accessibility information |
|---|---|---|---|---|
| Goedhart J, Von Stetten D, Noirclerc-Savoye M, Lelimousin M, Joosen L, Hink MA, Van Weeren L, Gadella TWJ, Royant A | 2012 | X-ray structure of the cyan fluorescent protein mTurquoise (K206A mutant) | 2YE0; http://www.rcsb.org/pdb/explore/explore.do?structureId=2ye0 | Publicly available at the RCSB Protein Data Bank (http://www.rcsb.org/). |
| Miyazono K, Miyakawa T, Sawano Y, Kubota K, Kang HJ, Asano A, Miyauchi Y, Takahashi M, Zhi Y, Fujita Y, Yoshida T, Kodaira K, Yamaguchi-Shinozaki K, Tanokura M | 2009 | Crystal structure of (+)-ABA-bound PYL1 in complex with ABI1 | 3JRQ; http://www.rcsb.org/pdb/explore/explore.do?structureId=3jrq | Publicly available at the RCSB Protein Data Bank (http://www.rcsb.org/). |
| Rekas A, Alattia JR, Nagai T, Miyawaki A, Ikura M | 2002 | Crystal structure of a yellow fluorescent protein with improved maturation and reduce environmental sensitivity | 1MYW; http://www.rcsb.org/pdb/explore/explore.do?structureId=1myw | Publicly available at the RCSB Protein Data Bank (http://www.rcsb.org/). |
| Nishimura N, Hitomi K, Arvai AS, Rambo RP, Hitomi C, Cutler SR, Schroeder JI, Getzoff ED | 2009 | Crystal structure of dimeric abscisic acid (ABA) receptor pyrabactin resistance 1 (PYR1) with ABA-bound closed-lid and ABA-free open-lid subunits | 3K3K; http://www.rcsb.org/pdb/explore/explore.do?structureId=3k3k | Publicly available at the RCSB Protein Data Bank (http://www.rcsb.org/). |

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
