## [Decision Letter]

Thank you for sending your work entitled “Genetically encoded reporters for the direct visualization of abscisic acid distribution and transport in Arabidopsis” for consideration at *eLife*. Your article has been favorably evaluated by a Senior editor, Detlef Weigel, and 3 reviewers, one of whom, Richard Amasino, is a member of our Board of Reviewing Editors.

Your paper represents an important technical advance and we will consider publication if you can address the following issues. Two of the issues require further experimentation.

1) Determine whether transgenic lines expressing the sensors are able to report endogenous changes in ABA concentration. This experiment is not overly burdensome as the transgenic lines are already in place and as a reviewer notes “mannitol treatment of seedlings would trigger endogenous ABA synthesis and could be used a fast/simple experiment to prove that the sensors are useful for measuring changes in endogenous ABA content.”

2) Determine whether transgenic lines expressing the sensors exhibit altered sensitivity to exogenous ABA. Because the sensors bind to ABA and possibly to other endogenous proteins required for ABA signal transduction, there is the issue of whether or not plants expressing the sensors exhibit altered ABA sensitivity. Again, this experiment is not overly burdensome; for example, straightforward growth inhibition assays could be used. As a reviewer notes “I don't think that the lines need to have 100 % wild type ABA sensitivity to be useful. It would be ideal if this is the case, but the key point is to know their inherent properties and limitations and the caveats that come along with using them.”

Other issues:

3) The authors show a very clear pH sensitivity titration curve showing the pH independence of the sensor above pH 7.0. Some discussion of expected pH changes in response to ABA would be useful to place this pH dependency in context for the usefulness of the sensor.

4) In Figure 3 it looks like both the FRET acceptor and donor signal increase after addition of ABA in the guard cell. You would expect the donor signal to rise and acceptor to fall as ABA levels increase. Does this mean there is significant synthesis of the sensor over the time period of the experiment?

5) There are quite a few papers that have attempted to estimate ABA levels in single cell types, for example, a manuscript from the Weiler lab made estimates for guard cells after stress (27). I think the older literature on this topic should be cited and discussed in the Introduction and Discussion.

---

## [Author Response]

*1) Determine whether transgenic lines expressing the sensors are able to report endogenous changes in ABA concentration. This experiment is not overly burdensome as the transgenic lines are already in place and as a reviewer notes “mannitol treatment of seedlings would trigger endogenous ABA synthesis and could be used a fast/simple experiment to prove that the sensors are useful for measuring changes in endogenous ABA*
*content.”*

Based on recent studies, ABA concentrations increase in roots and shoots several hours after stress treatments (31; 22). Older studies, using manually dissected guard cells, measured ABA concentration increases in guard cells 15 min after passive dehydration of leaves (28). ABAleon2.1 detected endogenous ABA concentration increases in guard cells 15 min after a drop in humidity (Figures 4 and 9) and 4 h after 100 mM NaCl treatment (Figure 9). Interestingly, 4 h of 300 mM sorbitol treatment did not induce ABA increases in guard cells (Figure 9). When five day-old seedlings were treated for 6 h, both NaCl and sorbitol induced ABA concentration increases in roots (Figure 9). These experiments are now described and discussed in the manuscript (Abstract; Results; third paragraph of the Discussion section entitled “Analyses of ABA concentration changes and long-distance ABA transport in Arabidopsis”).

*2) Determine whether transgenic lines expressing the sensors exhibit altered sensitivity to exogenous ABA. Because the sensors bind to ABA and possibly to other endogenous proteins required for ABA signal transduction, there is the issue of whether or not plants expressing the sensors exhibit altered ABA sensitivity. Again, this experiment is not overly burdensome; for example, straightforward growth inhibition assays could be used. As a reviewer notes “I don't think that the lines need to have 100 % wild type ABA sensitivity to be useful. It would be ideal if this is the case, but the key point is to know their inherent properties and limitations and the caveats that come along with*
*using them.”*

We thank the reviewers for suggesting these experiments. Phenotypical analyses have been performed with two ABAleon2.1 lines (line 3 and line 10) that express ABAleon2.1 at different levels (Figure 5). These lines were compared to Col-0 wild type, YFP-PYR1 and *abi1-3*/YFP-ABI1 (59) over-expression lines in ABA sensitivity assays (Figure 5). In ABA-induced stomatal closure assays both ABAleon2.1 lines exhibited responses to ABA, which were comparable to Col-0 wild type plants (Figure 5). However, both ABAleon2.1 lines exhibited a reduced ABA sensitivity in seed germination, cotyledon expansion and seedling growth assays (Figure 5). Interestingly, the degree of ABA sensitivity in both lines correlated with ABAleon2.1 expression levels, which were determined by quantitative fluorescence microscopy (Figure 5). Compared to *abi1-3*/YFP-ABI1, the ABA sensitivity of the ABAleon2.1 plants was similar in seedling growth assays (Figure 5), but less affected in seed germination and cotyledon expansion assays (Figure 5,C,F-H). The higher fluorescence intensity of both ABAleon2.1 lines compared to *abi1-3*/YFP-ABI1 indicates that ABAleon2.1 is much more highly expressed than YFP-ABI1. On the basis that ABAleon2.1 does not exhibit phosphatase activity *in vitro* (Figure 1) and that the degree of ABA sensitivity correlated with ABAleon2.1 expression levels, the reduced ABA sensitivity of ABAleon2.1 plants (Figure 5) might result from ABAleon2.1 mediated scavenging of physiologically relevant ABA concentrations in certain tissues. We believe that any method has its limitations and that it is helpful to analyze these, as now, in this study. The relevance, limitation and possible mechanism of reduced ABA sensitivity of ABAleon2.1 are now described and discussed in the manuscript (Abstract; Results section entitled “Arabidopsis plants expressing ABAleon2.1 are ABA hyposensitive; final paragraph of Discussion section entitled “Design of ABAleon reporters”).

Although we have not exchanged manuscripts with the Frommer laboratory, brief communications inform us that their reporters are designed differently, have a different dynamic range, and a complementary ABA concentration detection range. Based on these differences, the two classes of ABA reporters developed and analyzed by our respective laboratories could be utilized for different aspects of ABA biology and could be complementary.

*Other*
*issues:*

*3) The authors show a very clear pH sensitivity titration curve showing the pH independence of the sensor above pH 7.0. Some discussion of expected pH changes in response to ABA would be useful to place this pH dependency in context for the usefulness of the sensor*.

ABA is known to induce cytoplasmic alkalinization of guard cells. We now discuss previous literature showing this and discuss ABAleon stability in this pH range, as suggested (fifth paragraph of the Discussion section entitled “Analyses of ABA concentration changes and long-distance ABA transport in Arabidopsis).

*4) In*
Figure 3
*it looks like both the FRET acceptor and donor signal increase after addition of ABA in the guard cell. You would expect the donor signal to rise and acceptor to fall as ABA levels increase. Does this mean there is significant synthesis of the sensor over the time period of the*
*experiment?*

Examination of the imaging data showed, that the increase of acceptor emission in the guard cell analyses in Figure 3 was caused by a slight sample drift. Although it is known that ABA induces expression of certain PP2Cs and represses expression of certain PYR/PYL/RCARs (Leonhardt et al., 2004; Santiago et al., 2009; Szostkiewicz et al., 2010), an ABA dependent expression of ABAleon2.1 is rather unlikely. ABAleon2.1 was expressed under the control of the pUBQ10 promoter, which is ABA independent (http://bar.utoronto.ca/efp/cgi-bin/efpWeb.cgi; Winter et al., 2007). It is not expected, that significant changes in ABAleon2.1 protein levels occur within the time period of the experiments in Figure 3. In the case of *de novo* protein synthesis, newly synthesized ABAleon2.1 proteins would not affect the ratiometric measurements in the timeframe of the experiments in Figure 3, because the maturation time of mTurquoise is > 1 h (24). We have added text to the figure caption noting that a slight sample drift contributed to fluorescence signals in Figure 3.

*5) There are quite a few papers that have attempted to estimate ABA levels in single cell types, for example, a manuscript from the Weiler lab made estimates for guard cells after stress (*[27]*). I think the older literature on this topic should be cited and discussed in the Introduction and Discussion*.

Thank you for this comment. This section has been included in the Introduction: “In response to water limitations ABA concentrations increase (27; 28; 11; 18; 31; 22) and decrease upon stress relief (28; 17).”

This section has been included in the Discussion: “In *Vicia faba* guard cells ABA concentrations were ∼ 0.7 fg/cell pair in unstressed and ∼ 17.7 fg/cell pair in stressed guard cells (27). ABA concentrations in stressed guard cells were estimated to be in the range of ∼15 µM (27; 28). Extrapolating from these values, unstressed guard cell ABA concentration would be ∼ 500 nM. Such approximations would be consistent with the partial saturation and reduced response of ABAleon2.1 in guard cells (Figure 3) and with strong expression of the ABA-induced reporter pRAB18-GFP (Figure 2—figure supplement 1).”

In addition to the requested points, we have included a new Figure (Figure 8), in which ABA-induced responses of low-affinity ABAleons (ABAleon2.13, ABAleon2.14 and ABAleon2.15) were investigated in the root maturation zone of transgenic Arabidopsis plants. These analyses demonstrate the utility of ABAleon2.15 (K’_d_ ∼ 600 nM), which could complement the high affinity ABAleon2.1 (K’_d_ ∼ 100 nM) (Results section entitled “Low affinity ABAleon2.15 reports ABA uptake in roots”).